# Sparse Bayesian Deep Functional Learning with Structured Region Selection

Xiaoxian Zhu [1]  Yingmeng Li [1]  Shuangge Ma [2]  Mengyun Wu [1]

## Abstract

In modern applications such as ECG monitoring, neuroimaging, wearable sensing, and industrial equipment diagnostics, complex and continuously structured data are ubiquitous, presenting both challenges and opportunities for functional data analysis. However, existing methods face a critical trade-off: conventional functional models are limited by linearity, whereas deep learning approaches lack interpretable region selection for sparse effects. To bridge these gaps, we propose a sparse Bayesian functional deep neural network (sBayFDNN). It learns adaptive functional embeddings through a deep Bayesian architecture to capture complex nonlinear relationships, while a structured prior enables interpretable, region-wise selection of influential domains with quantified uncertainty. Theoretically, we establish rigorous approximation error bounds, posterior consistency, and region selection consistency. These results provide the first theoretical guarantees for a Bayesian deep functional model, ensuring its reliability and statistical rigor. Empirically, comprehensive simulations and real-world studies confirm the effectiveness and superiority of sBayFDNN. Crucially, sBayFDNN excels in recognizing intricate dependencies for accurate predictions and more precisely identifies functionally meaningful regions, capabilities fundamentally beyond existing approaches.

## 1. Introduction

With rapid technological advances, diverse fields—from healthcare to neuroscience—are generating increasingly complex data that exhibit inherent structure and continuity over time or space. For instance, electrocardiogram (ECG) recordings are analyzed as functional curves to elucidate cardiac morphology and identify pathological patterns, offering insights into cardiovascular health (Pang et al., 2023). Similarly, neuroimaging data are modeled as spatiotemporal fields to capture brain dynamics constrained by anatomical geometry, advancing the study of neural functions and disorders (Tsai et al., 2024). Such data are naturally viewed as realizations of underlying functions and are suited for analysis within functional data analysis (FDA), which uses their smoothness and structure to model temporal or spatial patterns. The analysis of these functional observations is important, particularly in human health studies, where interpreting such signals can inform diagnosis, monitoring, and treatment.

In FDA, supervised learning with functional predictors is fundamentally important yet faces two major challenges. First, the relationship between functional predictors and responses is often intrinsically complex and nonlinear. Ignoring such nonlinearity risks significant model misspecification, potentially leading to poor predictive performance in real-world applications. Second, and more critically, many applications exhibit local sparsity or region-specific effects. This means the predictive relationship is not globally active across the entire function domain; instead, it is concentrated within one or a few specific, contiguous subregions (e.g., time intervals or wavelength bands) where the functional coefficient is nonzero, while being negligible or zero elsewhere. Failure to account for this structure may introduce substantial noise and degrade both interpretability and estimation efficiency.

These two challenges are commonly intertwined in practice. For example, in bedside monitoring, physiological waveforms such as ECG—driven by intricate and dynamic pathophysiological states—not only manifest highly complex and nonlinear patterns but also frequently exhibit local sparsity (Moor et al., 2023). Specifically, clinically actionable information is not distributed uniformly but is concentrated within physiologically meaningful regions of the waveform, such as the QRS complex for arrhythmia analysis or the ST segment for ischemia detection. Signals from other intervals are often non-informative. This makes interpretable region selection a core enabling step, as it precisely targets the localized subdomains where complex nonlinear relationships are most active and meaningful, thereby forming the

[1]School of Statistics and Data Science, Shanghai University of Finance and Economics [2]Department of Biostatistics, Yale School of Public Health. Correspondence to: Mengyun Wu <wu.mengyun@mail.shufe.edu.cn>.

*Proceedings of the 43rd International Conference on Machine Learning*, Seoul, South Korea. PMLR 306, 2026. Copyright 2026 by the author(s).

foundation for efficient and robust functional models.

Supervised learning with functional data has been extensively studied. Classical functional linear regression (FLR) estimates smooth coefficient functions but lacks inherent region selection—a limitation partially mitigated by later sparse FLR variants, though linearity constraints remain. Nonlinear functional models offer flexibility yet often depend on prespecified structures, limiting their capacity to capture highly complex relationships. While deep neural networks (DNNs) provide strong approximation power and have been adapted to functional settings, existing DNN-based methods focus predominantly on prediction and do not incorporate interpretable, structured region selection on the functional domain. Moreover, common DNN feature selection techniques are designed for scalar inputs and fail to leverage the continuous nature of functional data. These gaps underscore the need for a DNN framework that jointly handles nonlinear complexity and performs interpretable region selection in functional settings.

We propose a sparse Bayesian deep neural network (sBayFDNN) that integrates functional embedding learning with a Bayesian DNN architecture to perform interpretable region-wise selection for functional predictors. The model captures flexible nonlinear function-to-scalar relationships while providing posterior inference for uncertainty in region selection. Our main contributions are:

- A Bayesian Functional DNN Framework with Uncertainty Quantification: We introduce a model that embeds functional predictors through a deep Bayesian network to capture complex nonlinear associations with scalar responses. The framework delivers not only point estimates but also posterior uncertainty for the selected regions, overcoming the rigidity of traditional functional regression.

- Interpretable Region Selection via Structured Sparsity: A sparse prior is imposed on the first hidden layer to perform region-wise functional selection. This significantly enhances model interpretability while mitigating the black-box nature of conventional DNNs, yielding actionable insights into influential domains.

- Theoretical Guarantees: We establish rigorous theoretical results, including approximation error bounds, posterior consistency, and region selection consistency. These provide the first theoretical foundation for a Bayesian deep functional model, thereby opening a new venue for statistically rigorous and interpretable functional deep learning.

- Empirical Validation: Comprehensive simulations and diverse real-world studies confirm that sBayFDNN outperforms existing methods in both prediction accuracy

and region identification, especially in challenging scenarios where conventional models are misspecified or inflexible, demonstrating its practical superiority.

## 2. Related Works

### 2.1. Classical Functional Regression Methods

Classical functional regression methods, represented by the functional linear model and its extensions, establish relationships between functional predictors and scalar responses using techniques such as B-splines (Cardot et al., 2003), reproducing kernel Hilbert spaces (Crambes et al., 2009), and smoothing regularization (Cai & Yuan, 2012). To enhance interpretability, sparse penalization methods have been introduced for local region selection, including sparse functional linear regression (Lee & Park, 2012), sparse functional data embeddings with localized functional principal component analysis (Chen & Lei, 2015), as well as smooth locally sparse estimators (Lin et al., 2017; Belli, 2022) and their Bayesian counterparts (Zhu et al., 2025). To move beyond linearity, nonlinear methods such as functional generalized additive models (McLean et al., 2014) and functional single-index models have been proposed (Jiang et al., 2020). These nonlinear models are further combined with the $L_0$ norm (Chamon et al., 2020) and the functional SCAD penalty (Nie et al., 2023) to achieve local sparsity.

However, these approaches face key limitations: linear models cannot adequately capture complex nonlinear relationships, while existing nonlinear methods rely on pre-defined kernels or bases, limiting their flexibility and capacity to model intricate dependencies directly from data.

### 2.2. Deep Learning for Functional Data

Deep learning for functional data has advanced in recent years, with methods primarily aimed at enhancing representation learning and predictive performance. Early approaches introduced adaptive basis layers to learn data-driven functional representations within DNNs (Yao et al., 2021), and were followed by DNNs specifically designed for functional inputs (Thind et al., 2023) and classifiers built on functional principal components (Wang et al., 2023). Autoencoder architectures have also been adapted for functional latent representation learning (Wu et al., 2024).

Despite their predictive power, these methods often operate as "black boxes", lacking explicit mechanisms for region selection on the functional domain. This limits their utility in scientific applications where identifying which specific regions of the data drive predictions is essential.

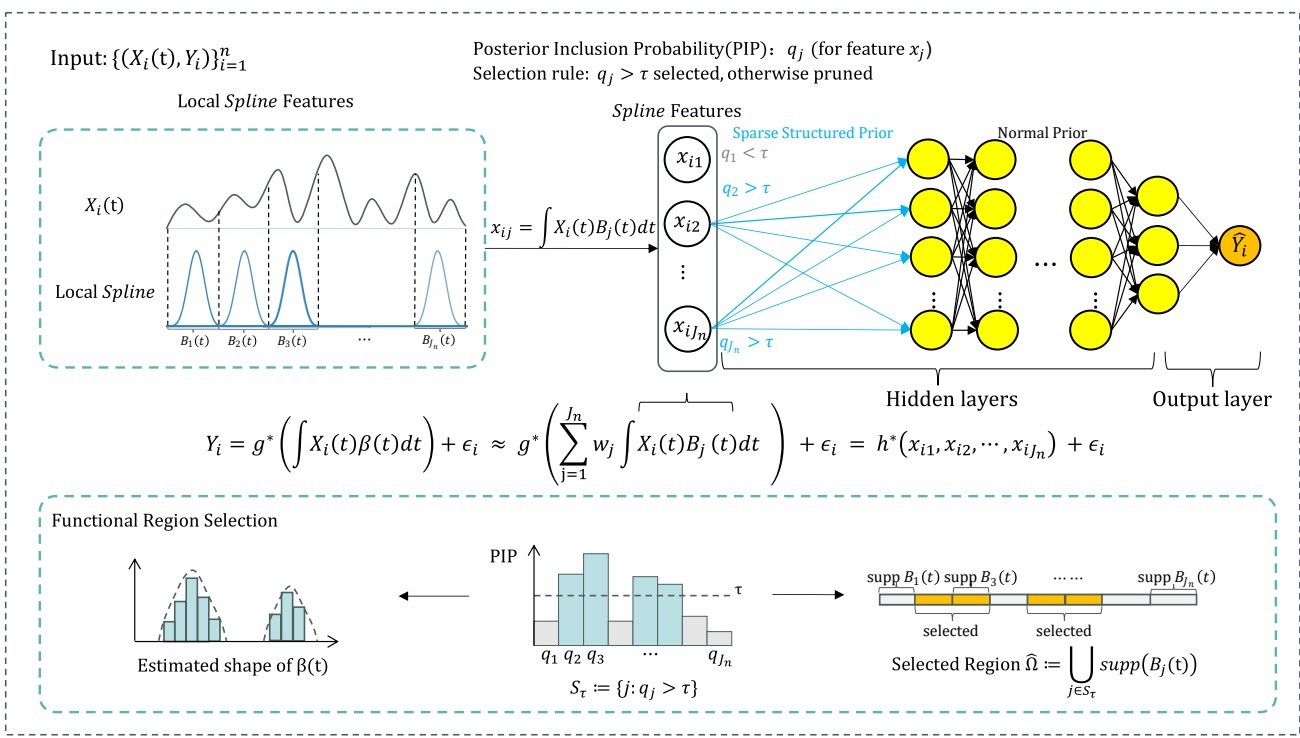

*Figure 1.* Workflow of sBayFDNN. The functional predictor $X_i(t)$ is projected onto locally supported B-spline bases to form spline features $\boldsymbol{x}_i = (x_{i1}, \ldots, x_{i,J_n})^\top$. A DNN with a spike-and-slab prior on the first-layer weight columns yields feature-wise posterior inclusion probabilities (PIPs) $q_j$, which are thresholded to select spline features and mapped back to the function domain to produce an estimated active region $\widehat{\Omega}$.

## 2.3. Sparsity and Selection in Neural Networks

A parallel line of research incorporates sparsity into neural networks for vector-valued inputs. This includes frameworks performing variable selection via group sparsity penalties on the first hidden layer (Dinh & Ho, 2020; Chu et al., 2023; Luo & Halabi, 2025), architectures employing sparsity-inducing mechanisms on linear input layers (Chen et al., 2021; Atashgahi et al., 2023), and residual layers used to achieve feature-wise sparsity (Lemhadri et al., 2021; Fan & Waldmann, 2025). Furthermore, Sun et al. (2022) proposed a Bayesian framework to learn sparse DNNs through connection pruning, establishing posterior consistency but not traditional input-level variable selection. Beyond variable selection, similar sparse DNN techniques have also been developed for applications in other fields, such as biological or social network reconstruction (Fan et al., 2026; Yang et al., 2026).

While these techniques enhance interpretability for vector data, they are not designed to exploit the continuous, structured nature of functional inputs. Consequently, a significant gap remains for a method that integrates nonlinear deep learning with structured, interpretable region selection and uncertainty quantification specifically for functional data.

## 3. Methodology

Let $X_1(t), \ldots, X_n(t)$ be $n$ independent functional covariates defined on the closed interval $\mathcal{T} \subset \mathbb{R}$. For theoretical and computational purposes, $\mathcal{T}$ is standardized to $[0, 1]$. Let $Y_i$ denote a scalar response. We consider a functional single-index model (see Figure 1 for a schematic overview):

$$Y_i = g^* \left( \int_{\mathcal{T}} X_i(t)\, \beta(t)\, dt \right) + \varepsilon_i, \qquad (1)$$

where $\varepsilon_i \sim \mathcal{N}(0, \sigma_\varepsilon^2)$, $\beta(\cdot)$ is an unknown coefficient function, and $g^*(\cdot)$ is an unknown nonlinear function. Without loss of generality, we assume the domain of $g^*(\cdot)$ is $[0, 1]$; this can be achieved by suitably rescaling $\int_{\mathcal{T}} X_i(t)\beta(t)dt$ and absorbing the scaling into the definition of $g^*$. For notational simplicity, we retain the original notation in the subsequent exposition.

The function $g^*(\cdot)$ allows for a flexible, potentially nonlinear relationship between the scalar response and the functional covariate through the single-index projection. Motivated by applications where the association is driven by only a few subregions of $\mathcal{T}$ (e.g., a small number of time intervals), we assume that $\beta(\cdot)$ is effectively localized and aim to identify the corresponding active regions.

To circumvent the infinite-dimensionality of $\beta(\cdot)$, we ap-

proximate it using a truncated B-spline basis expansion:

$$\beta_{J_n}(t) = \sum_{j=1}^{J_n} w_{J_n,j} B_j(t), \tag{2}$$

which yields the following finite-dimensional representation:

$$Y_i \approx g^*\big(\boldsymbol{\eta}(X_i)^\top \boldsymbol{w}_{J_n}\big) + \varepsilon_i = h^*\big(\boldsymbol{\eta}(X_i)\big) + \varepsilon_i, \tag{3}$$

by replacing $\beta(\cdot)$ with $\beta_{J_n}(\cdot)$, where $\int_{\mathcal{T}} X_i(t)\beta_{J_n}(t)\,dt = \boldsymbol{\eta}(X_i)^\top \boldsymbol{w}_{J_n}$. Here, $\{B_j(\cdot)\}_{j=1}^{J_n}$ denotes a collection of $J_n$ B-spline basis functions, and $\boldsymbol{w}_{J_n} := (w_{J_n,1}, \ldots, w_{J_n,J_n})^\top \in \mathbb{R}^{J_n}$ is the corresponding vector of coefficients. The spline features $\boldsymbol{\eta}(X_i) = (x_{i1}, \ldots, x_{iJ_n})^\top \in \mathbb{R}^{J_n}$, where $x_{ij} := \int_{\mathcal{T}} X_i(t)\,B_j(t)\,dt$. We further introduce $h^*(\cdot)$, which absorbs the B-spline coefficient vector $\boldsymbol{w}_{J_n}$, thus reducing the problem to learning a finite-dimensional function.

Next, we aim to estimate $h^*(\cdot)$ using a DNN $F_{\boldsymbol{\theta}}$ that takes $\boldsymbol{\eta}(X_i) \in \mathbb{R}^{J_n}$ as input. The network can be written as

$$F_{\boldsymbol{\theta}} = A_{H_n} \circ \sigma \circ A_{H_n-1} \circ \cdots \circ \sigma \circ A_1. \tag{4}$$

In (4), we consider a feedforward DNN with $H_n - 1$ hidden layers and width $L_h$ at layer $h$, where $L_0 = J_n$ and $L_{H_n} = 1$, with a common activation $\sigma(\cdot)$ in the hidden layers and a linear output layer. We consider the ReLU activation in our study. Let $\boldsymbol{\theta} = \{(\boldsymbol{W}_h, \boldsymbol{b}_h)\}_{h=1}^{H_n}$, where $\boldsymbol{W}_h \in \mathbb{R}^{L_h \times L_{h-1}}$ and $\boldsymbol{b}_h \in \mathbb{R}^{L_h}$ denote the weights and biases. The affine map is defined as $A_h(\boldsymbol{u}) := \boldsymbol{W}_h \boldsymbol{u} + \boldsymbol{b}_h$.

Here, the DNN $F_{\boldsymbol{\theta}}$ directly estimates the composite function $h^*(x_{i1}, \ldots, x_{i,J_n}) = g^*\big(\sum_{j=1}^{J_n} w_{J_n,j} x_{i,j}\big)$, which is the transformation of the single-index model after spline expansion. This perspective makes clear that the single-index structure is a special case of the DNN model class. Importantly, our goal is not to explicitly recover the exact single-index architecture; rather, we aim to find a DNN $F_{\boldsymbol{\theta}}$ within this class that accurately estimates the conditional expectation $E(Y_i \mid X_i(t))$ under the true single-index model, which is a standard approach in nonparametric statistical learning.

In (3), since the B-spline basis is locally supported, the localization of $\beta(\cdot)$ over $\mathcal{T}$ is naturally reflected in the sparsity pattern of $\boldsymbol{w}_{J_n}$ in (2). Therefore, we consider the setting where $w_{J_n}$ is sparse, so that $\boldsymbol{\eta}(X_i)^\top \boldsymbol{w}_{J_n} = \boldsymbol{\eta}_{S_{J_n}}(X_i)^\top \boldsymbol{w}_{S_{J_n}}$, where $S_{J_n} \subset \{1, \ldots, J_n\}$ denotes an unknown support set, $\boldsymbol{\eta}_{S_{J_n}}(X_i)$ and $\boldsymbol{w}_{S_{J_n}}$ denote the corresponding subvectors. With (4), since $\boldsymbol{w}_{J_n}$ is absorbed into $h^*(\cdot)$, the sparsity of $\boldsymbol{w}_{J_n}$ naturally translates into sparsity of the first-layer weight matrix $\boldsymbol{W}_1 \in \mathbb{R}^{L_1 \times J_n}$. Specifically, denoting by $\boldsymbol{W}_{1,*j} \in \mathbb{R}^{L_1}$ the $j$-th column of $\boldsymbol{W}_1$, we have that if $||\boldsymbol{W}_{1,*j}||_2 \neq 0$, then $j \in S_{J_n}$.

Based on (3) and (4), we propose a sparse Bayesian functional DNN (sBayFDNN) framework, employing the following prior distributions for the model parameters:

for $j \in \{1, \ldots, J_n\}$,

$$\boldsymbol{W}_{1,*j} \mid \gamma_j \sim \left[\mathcal{N}(0, \sigma_{1,n}^2 \boldsymbol{I})\right]^{\gamma_j} \left[\mathcal{N}(0, \sigma_{0,n}^2 \boldsymbol{I})\right]^{1-\gamma_j} \tag{5}$$

$$\pi(\gamma_j) \sim \mathrm{Bern}(\lambda_n), \tag{6}$$

$$W_{h,ab} \sim \mathcal{N}(0, \sigma^2), \ \forall h \in \{2, \ldots, H_n\}, \tag{7}$$

$$b_{h,a} \sim \mathcal{N}(0, \sigma^2), \ \forall h \in \{1, \ldots, H_n\}. \tag{8}$$

Here, $\mathcal{N}(c, d)$ denotes the normal distribution with mean $c$ and variance (covariance) $d$, and $\mathrm{Bern}(\lambda_n)$ denotes the Bernoulli distribution with success probability $\lambda_n \in (0, 1)$. The symbol $\boldsymbol{I}$ represents an identity matrix of appropriate dimensions. The entry $W_{h,ab}$ corresponds to the $(a, b)$-th element of the matrix $\boldsymbol{W}_h$, where $a \in 1, \ldots, L_h$ and $b \in 1, \ldots, L_{h-1}$. Each $\gamma_j$ (for $j \in 1, \ldots, J_n$) is a binary indicator. We set $\sigma_{1,n}^2 > \sigma_{0,n}^2 > 0$, with $\sigma_{0,n}^2$ taken to be a small positive value, and let $\sigma^2 > 0$.

As introduced above, distinct priors are specified for the first layer and the subsequent layers. To enable functional region selection in the spline-feature domain, we impose a group-wise continuous spike-and-slab prior on each $\boldsymbol{W}_{1,*j}$ for $j = 1, \ldots, J_n$. When $\gamma_j = 0$, the spike variance $\sigma_{0,n}^2$ strongly shrinks all entries of $\boldsymbol{W}_{1,*j}$ toward zero with high probability; when $\gamma_j = 1$, the slab variance $\sigma_{1,n}^2$ permits the entries to take appreciable nonzero values, thereby allowing the corresponding spline-basis feature to contribute to the functional representation. For the remaining layers, weights and biases are assigned entrywise independent Gaussian priors $\mathcal{N}(0, \sigma^2)$, which preserves the flexibility needed in the hidden layers to learn complex, nonlinear patterns based on the features identified by the first hidden layer.

## 4. Optimization-based Bayesian Inference

The proposed Bayesian inference proceeds via three steps: (i) compute a (local) posterior mode (MAP) of the network parameters under the marginal prior; (ii) derive feature-level posterior inclusion probabilities from the first-layer; and (iii) map the selected spline features to an estimated active region. The detailed procedure is described below, while hyperparameter settings, sensitivity analyses, and runtime results are provided in Appendix A.

### 4.1. MAP Fitting

For $\boldsymbol{\theta}$, we optimize the marginal posterior in which the inclusion indicators $\boldsymbol{\gamma}$ are integrated out, and denote the resulting marginal prior by $\pi(\boldsymbol{\theta})$. Then, we compute a MAP

by minimizing the negative log-posterior objective

$$\hat{\boldsymbol{\theta}} \in \arg\min_{\boldsymbol{\theta}} \; \frac{1}{2\sigma_\varepsilon^2} \sum_{i=1}^n \Big( Y_i - F_{\boldsymbol{\theta}}(\boldsymbol{\eta}(X_i)) \Big)^2 \; - \; \log \pi(\boldsymbol{\theta}),$$
(9)

where $F_{\boldsymbol{\theta}}$ is the $H_n$-layer feedforward network defined in (4). Optimization is carried out using stochastic gradient methods (see Appendix A for the detailed algorithm).

### 4.2. MAP plug-in Posterior Inclusion Probabilities

Denote $A_{1,n} = \lambda_n(\sigma_{1,n}^2)^{-L_1/2}$ and $A_{0,n} = (1 - \lambda_n)(\sigma_{0,n}^2)^{-L_1/2}$. Then, for each feature index $j$, applying Bayes' rule to (5) and (6) gives the conditional posterior inclusion probability (PIP): $\Pr(\gamma_j = 1 \mid \boldsymbol{W}_{1,*j}) =$

$$\frac{A_{1,n}\exp\left(-\frac{\|\boldsymbol{W}_{1,*j}\|_2^2}{2\sigma_{1,n}^2}\right)}{A_{1,n}\exp\left(-\frac{\|\boldsymbol{W}_{1,*j}\|_2^2}{2\sigma_{1,n}^2}\right)+A_{0,n}\exp\left(-\frac{\|\boldsymbol{W}_{1,*j}\|_2^2}{2\sigma_{0,n}^2}\right)}.$$

We compute plug-in PIP $\hat{q}_j := \Pr(\gamma_j = 1 \mid \hat{\boldsymbol{W}}_{1,*j})$ based on the MAP estimate $\hat{\boldsymbol{\theta}}$ and select spline features by thresholding: $\hat{S}_\tau := \{j : \hat{q}_j > \tau\}$ with default $\tau = 1/2$. Since $\hat{q}_j$ is monotone in $\|\hat{\boldsymbol{W}}_{1,*j}\|_2^2$, this rule is equivalent to thresholding the first-layer column norms.

### 4.3. Mapping Selected Features to an Active Region on $\mathcal{T}$

Given $\hat{S}_\tau \subset \{1, \ldots, J_n\}$, we map the selected spline features back to the functional domain using the local support of B-splines. Let $\mathrm{supp}(B_j(t)) \subset \mathcal{T}$ denote the support of the $j$th basis function. We define the estimated active region as $\hat{\Omega} := \bigcup_{j \in \hat{S}_\tau} \mathrm{supp}(B_j)$, and represent $\hat{\Omega}$ as a union of disjoint subintervals by merging adjacent components.

### 4.4. Practical Justification of MAP-based PIPs

While full Bayesian sampling is more common for uncertainty quantification, MAP-based plug-in PIPs are also well established in sparse Bayesian modeling and sparse deep learning (Gan et al., 2019; Ročková, 2018; Ročková & George, 2018; Yang et al., 2021; Sun et al., 2022), where they offer both theoretical guarantees and computational efficiency. In our setting, we therefore treat the resulting PIPs as approximate posterior selection scores rather than exact posterior marginals. We further examine their empirical stability across random initializations; the corresponding results are reported in Appendix A.6.

## 5. Theoretical Properties

Let $\beta^*$ and $\beta_{J_n}^*$ be the true coefficient function and its truncated counterpart, respectively. Let $\boldsymbol{\omega}_{J_n}^* \in \mathbb{R}^{J_n}$ be the vector of true basis coefficients for $\beta_{J_n}^*$, $S_{J_n}^* = \mathrm{supp}(\boldsymbol{\omega}_{J_n}^*)$ its support, and $s_n = |S_{J_n}^*|$ the sparsity level. Define

$e_j^* = \mathbb{I}(j \in S_{J_n}^*)$ as the true binary selection indicator and $\Omega^* \subset \mathcal{T}$ as the true non-zero region of $\beta^*$. Denote $\Omega^*(\kappa) = \{t \in \mathcal{T} : |\beta^*(t)| > \kappa\}$ as the strong-signal region and $\Omega^*(\kappa)^c$ as its complement. Denote $a_n \lesssim b_n$ if $a_n \leq C b_n$ for some constant $C > 0$, and $a_n \asymp b_n$ if $c\, b_n \leq a_n \leq C b_n$ for constants $0 < c < C$.

Denote $\mu^*(X) = g^* \left( \int_{\mathcal{T}} X(t)\beta^*(t)dt \right)$ as the true mean function and $\mu_{\boldsymbol{\theta}}(X) := F_{\boldsymbol{\theta}}(\boldsymbol{\eta}(X))$ as the network output defined in (4). Consider the class of fully-connected, $J_n$-input ReLU networks, denoted by $\mathcal{NN}_{J_n}(H_n, \bar{L}, E_n)$, with depth $H_n$, constant maximum hidden width $\bar{L} = \max_{2 \leq k \leq H_n} L_k$, and parameter bound $\|\boldsymbol{\theta}\|_\infty \leq E_n$. We define the column-wise support of the network parameters as $\mathrm{supp}_{\mathrm{col}}(\boldsymbol{\theta}) = \{j \in \{1, \cdots, J_n\} : \|\boldsymbol{W}_{1,*j}\|_2 > 0\}$. For any subset $T \subset \{1, \cdots, J_n\}$, the associated column-sparse network class is defined as $\mathcal{NN}_{J_n}^{\mathrm{col}}(T; H_n, \bar{L}, E_n) := \{F_{\boldsymbol{\theta}} \in \mathcal{NN}_{J_n}(H_n, \bar{L}, E_n) : \mathrm{supp}_{\mathrm{col}}(\boldsymbol{\theta}) = T\}$.

The following assumptions are imposed.

**Assumption 5.1.** $\sup_{X_i \in \mathcal{X}} \int_{\mathcal{T}} |X_i(t)|dt \leq C_X$, where $\mathcal{X}$ denotes the function space to which $X_i(t)$ belongs, and $C_X$ is a positive constant .

**Assumption 5.2.** $\beta^*(t) \in \mathcal{H}^{\alpha_\beta}([0,1])$ for some $\alpha_\beta > 0$, where $\mathcal{H}^\alpha(I)$ denotes the Hölder space of functions on an interval $I$ with smoothness order $\alpha$. $\Omega^*$ is a finite union of intervals and $\left|\Omega^*(\kappa_{J_n})^c\right| \to 0$, as $n \to \infty$, with $\kappa_{J_n} = c_\kappa C_\beta J_n^{-\alpha_\beta}$, where $c_\kappa > 4$ and $C_\beta$ are constants defined in Lemma B.1 (Appendix). Moreover, the strong-signal region has proportional size in the sense that $\frac{|\Omega^*(\kappa_{J_n})|}{|\Omega^*|} \geq c_\Omega$ for some constant $c_\Omega \in (0,1]$ and all sufficiently large $n$.

**Assumption 5.3.** $g^*(\cdot) \in \mathcal{H}^{\alpha_g}([0,1])$ for some constant $\alpha_g > 0$.

These are standard regularity conditions in functional data analysis (Cai & Yuan, 2012; Nie et al., 2023). Assumption 5.1 ensures the well-posedness of the problem by uniformly bounding the predictor trajectories. Assumption 5.2 imposes Hölder smoothness on $\beta^*$, guaranteeing its accurate spline approximation. We further assume the active region $\Omega^*$ is not overly complex, which excludes highly fragmented supports and aligns with typical domain-selection interpretations. Crucially, we require that the strong-signal region essentially covers $\Omega^*$ up to a negligible set; this acts as a minimum-signal condition in the functional setting, ensuring that the nonzero part of $\beta^*$ is detectable enough to translate basis selection into faithful support recovery on the domain. This minimum signal strength condition is standard in high dimensional variable and region selection literature. We further focus on the practically common proportional sparsity regime, where the strong-signal region retains a non-vanishing fraction of the active domain, which, under locally supported spline bases, implies that the number of active groups scales as $s_n \asymp J_n$. Assumption 5.3 similarly

enforces smoothness on the nonparametric link function, thereby supporting stable estimation.

All theoretical results are established for the DNN-based model $Y \sim p_{\mu_\theta}$ with $\mu_\theta(X) = F_\theta(\eta(X))$. The true data are generated from the single-index model $Y \sim p_{\mu^*}$ with $\mu^*(X) = g^*(\int X(t)\beta^*(t)dt)$. Theorems 5.4, 5.7, and 5.9 therefore analyze the approximation, posterior contraction, and selection consistency of the DNN estimator when the truth follows a single-index structure, demonstrating that the fitted DNN procedure is statistically valid even though the model class is larger.

**Theorem 5.4.** *Suppose that Assumptions 5.1–5.3 hold. Then:*

(i) *There exists a network parameter vector $\theta$ such that:*

$$F_\theta = F \in \mathcal{NN}_{J_n}^{\mathrm{col}}(S_{J_n}^*; H_n, \bar{L}, E_n),$$

(ii) *With $\alpha_1 = \min(\alpha_g, 1)$,*

$$\sup_{X \in \mathcal{X}} |\mu_\theta(X) - \mu^*(X)| \lesssim H_n^{-2\alpha_1} + J_n^{-\alpha_\beta \alpha_1}.$$

Theorem 5.4 establishes that the true sparse model structure is exactly representable within the proposed architecture (i) and provides a non-asymptotic error bound (ii). This bound guarantees that the approximation error decays to zero as the network depth $H_n$ and the number of spline bases $J_n$ grow, with the rates governed by the smoothness of both the functional coefficient $\beta(t)$ and the link function $g^*(\cdot)$.

**Assumption 5.5.** For some constants $\tau' > 0$ and $\alpha_\sigma > 0$:
$\lambda_n \lesssim \frac{1}{J_n\left[(n\bar{L})^{H_n}(J_n+1)L_1\right]^{\tau'}}$, $\frac{E_n^2}{H_n(\log n + \log \bar{L})} \lesssim \sigma_{1,n}^2 \lesssim$
$n^{\alpha_\sigma}$, and $\frac{E_n^2}{H_n(\log n + \log \bar{L})} \lesssim \sigma^2$.

**Assumption 5.6.** The DNN $F_\theta$ in (4) satisfies: $\bar{L} \asymp L_1 \asymp 1$, $H_n \asymp \min\{J_n^{\alpha_\beta/2}, s_n\}$ with $s_n \asymp J_n$, and $\|\theta\|_\infty \leq E_n$, where $E_n = n^{c_1}$ for some positive constant $c_1$.

Assumption 5.5 specifies the rates for key hyperparameters in the Bayesian framework, ensuring the prior is sufficiently diffuse to permit effective posterior contraction toward the true parameter, a standard requirement in Bayesian theory (Ghosal et al., 2000). Assumption 5.6 links the network depth $H_n$ to the sparsity level $s_n$ so that architectural growth respects the intrinsic sparse structure. Similar conditions are commonly adopted in related theoretical analyses.

Let $\Pi(A \mid D_n)$ denote the posterior probability of an event $A$ given the observed data $D_n = \{(X_i, Y_i)\}_{i=1}^n$. Let $p_{\mu^*}(\cdot \mid X)$ be the true conditional density of $Y$ given $X$, $p_{\mu_\theta}(\cdot \mid X)$ be the approximate density induced by the finite-dimensional representation (3) and the sparse DNN defined in (4), and $d(\cdot, \cdot)$ denote the Hellinger distance.

**Theorem 5.7.** *Suppose that Assumptions 5.1–5.6 hold and there exists an error sequence $\varepsilon_n^2$ satisfying $\varepsilon_n^2 \lesssim \frac{s_n \log(J_n/s_n)}{n} + \frac{s_n\left(H_n \log n + \log J_n\right)}{n} + \left(H_n^{-2\alpha_1} + J_n^{-\alpha_\beta\alpha_1}\right)^2$ such that $\sigma_{0,n}^2 \leq \tilde{M}_{n,1}(\varepsilon_n)$ and $\max\{\sigma^2, \sigma_{0,n}^2, \sigma_{1,n}^2\} \leq \tilde{M}_{n,2}(\varepsilon_n)$, where $\tilde{M}_{n,1}(\varepsilon_n)$ and $\tilde{M}_{n,2}(\varepsilon_n)$ are defined in (18) and (19) of Appendix. Then, for some constant $c > 0$, we have*

$$P\left[\Pi\{d(p_{\mu_\theta}, p_{\mu^*}) > 4\varepsilon_n \mid D_n\} \geq 2\exp(-c\, n\, \varepsilon_n^2)\right]$$
$$\leq 2\exp(-c\, n\, \varepsilon_n^2),$$

*and*

$$\mathbb{E}\left[\Pi\{d(p_{\mu_\theta}, p_{\mu^*}) > 4\varepsilon_n \mid D_n\}\right] \leq 4\exp(-2c\, n\, \varepsilon_n^2).$$

Theorem 5.7 establishes the posterior contraction rate $\varepsilon_n$ of sBayFDNN. This result implies that, with high probability, the posterior distribution concentrates around the true data-generating process at rate $\varepsilon_n$. Specifically, if $\alpha_\beta \leq 2$, then $H_n \asymp J_n^{\alpha_\beta/2}$, setting $J_n \asymp \left(\frac{n}{\log n}\right)^{\frac{1}{1+\alpha_\beta/2+2\alpha_\beta\alpha_1}}$ yields $\varepsilon_n^2 \asymp \left(\frac{\log n}{n}\right)^{\frac{2\alpha_\beta\alpha_1}{1+\alpha_\beta/2+2\alpha_\beta\alpha_1}}$. If $\alpha_\beta > 2$, then $H_n \asymp s_n$, taking $J_n \asymp \left(\frac{n}{\log n}\right)^{\frac{1}{2+4\alpha_1}}$ leads to $\varepsilon_n^2 \asymp \left(\frac{\log n}{n}\right)^{\frac{2\alpha_1}{1+2\alpha_1}}$.

Further, define the structural difference as

$$\rho_n(\varepsilon_n) := \max_{1\leq j \leq J_n} \mathbb{E}\left[|\gamma_j - e_j^*| \cdot \mathbb{I}\{\theta \notin A_n(\varepsilon_n)\} \,\Big|\, D_n\right],$$

where $\mathbb{E}(\cdot|D_n)$ is the conditional expectation, and $A_n(\varepsilon_n) = \left\{\theta: d(p_{\mu_\theta}, p_{\mu^*}) \geq \varepsilon_n\right\}$.

**Assumption 5.8.** $\rho_n(4\varepsilon_n) \to 0$ as $n \to \infty$ and $\varepsilon_n \to 0$.

Assumption 5.8 serves as an identifiability condition. It implies that as $n \to \infty$ and $\varepsilon_n \to 0$, any candidate model that is close to the true data-generating process in terms of Hellinger distance must asymptotically share the same underlying structure, thereby guaranteeing the consistent selection (Sun et al., 2022). This assumption should be understood as excluding correlation-equivalent sparse solutions that are nearly indistinguishable in predictive terms. Its role is not to assert that such ambiguity never arises in functional data, but rather to isolate a regime in which exact structural recovery is theoretically meaningful. When neighboring components of the functional predictor are highly correlated, the assumption becomes harder to verify, and the support recovered by the method may be better interpreted as a predictive sparse representative rather than a uniquely identifiable ground-truth structure.

**Theorem 5.9.** *Suppose Assumptions 5.1–5.8 hold. Then:*

(i) $\max_{1\leq j \leq J_n} |\hat{q}_j - e_j^*| \xrightarrow{P} 0$.

(ii) $P\left(S^*_{J_n} \subset \widehat{S}_\tau\right) \to 1$ *for any prespecified* $\tau \in (0,1)$.

(iii) $P\left(\widehat{S}_{1/2} = S^*_{J_n}\right) \to 1$.

(iv) $|\widehat{\Omega} \Delta \Omega^*| \xrightarrow{P} 0$.

Theorem 5.9 establishes the asymptotic consistency of spline feature selection based on MAP plug-in posterior probabilities, showing that the estimated selection probabilities converge to the true binary indicators. Furthermore, it guarantees the asymptotically exact recovery of the nonzero region in the continuous function domain from its discrete coefficient support, thereby achieving exact structural selection for the functional effect.

## 6. Experiments

We evaluate sBayFDNN on synthetic and real-world functional data against five competitors: (1) FNN, a spline-feature-based feedforward network using truncated basis expansion (Thind et al., 2023); (2) AdaFNN, which learns adaptive basis functions via auxiliary networks (Yao et al., 2021); (3) cFuSIM, a functional single-index method with localized regularization (Nie et al., 2023); (4) BFRS, a Bayesian functional region selector with neighborhood structure (Zhu et al., 2025); and (5) SLoS, a functional linear estimator with local sparsity via an fSCAD penalty (Lin et al., 2017). Predictive RMSE is reported for all methods; region-recovery metrics (Recall, Precision, F1) are provided only for sBayFDNN, cFuSIM, BFRS, and SLoS, as FNN and AdaFNN do not perform region selection.

### 6.1. Simulation Studies

Our simulation studies vary three key aspects: (1) the true coefficient function $\beta^*(t)$, which spans a single interior bump (Simple), a boundary bump (Medium), and a pair of narrow oscillating peaks (Complex); (2) the link function $g^*$, taken as linear, logistic, sinusoidal, or a composite nonlinear form; and (3) the response signal-to-noise ratio (SNR), set to 5 or 10. Functional covariates are generated from a truncated cosine basis, observed on a discrete grid with added measurement noise (see Appendix for full details).

Figure 2 summarizes region-recovery performance (F1 score) across all simulation scenarios (see Appendix for Recall and Precision). sBayFDNN consistently achieves high mean F1 with tight interquartile ranges, demonstrating robust and stable region identification across both linear and nonlinear regimes. Its advantage is more pronounced in more challenging scenarios with lower signal-to-noise ratios, as well as under harder localization regimes and complex nonlinear link functions $g^*$. By contrast, several baselines exhibit increased dispersion and more frequent low-F1 outcomes as moves away from central support or $g^*$

departs from linearity.

The distributions also suggest different selection behaviors across methods. For example, cFuSIM often returns broader active regions (capturing truly active intervals but at the cost of more false positives), whereas linear region-selection baselines deteriorate under strong nonlinearity due to model misspecification. In contrast, sBayFDNN maintains a better balance between recall and precision. Beyond aggregate metrics, we further evaluate uncertainty quantification. Taking the high-SNR, Medium-$\beta$, logistic-link scenario as an example, Figure 3 presents the estimated PIPs from sBayFDNN together with the normalized true coefficient function. The estimated PIPs vary with the signal strength of the true $\beta(t)$: regions with larger signal magnitude are assigned higher inclusion probabilities, while weaker or boundary regions receive more moderate probabilities. This suggests that the PIPs capture not only the location of the active regions but also the relative strength and detectability of the underlying signal.

Figure 4 shows that sBayFDNN achieves the lowest or near-lowest mean RMSE in most scenarios, with concentrated distributions, indicating that region interpretability does not come at the cost of predictive performance. Its advantage is most evident under nonlinear links, where flexible function approximation is crucial. Here, sBayFDNN substantially outperforms the spline-based FNN, matches the accuracy of AdaFNN with greater stability, and surpasses cFuSIM when nonlinearity is pronounced. Linear methods (BFRS, SLoS) remain competitive only under nearly linear settings, with performance degrading sharply as nonlinearity increases.

To further assess robustness beyond the main settings, we additionally examine low-SNR regimes, FGAM mean misspecification, and non-Gaussian heavy-tailed errors; these results are reported in Appendix C.2 and Appendix C.3. Together, these results affirm the sBayFDNN framework: structured sparsity in the first layer supports interpretable region identification with inherent uncertainty quantification, while subsequent deep layers furnish the flexibility needed for accurate nonlinear prediction.

### 6.2. Real Data Analysis

We evaluate our method on four benchmark datasets: ECG, Tecator, Bike rental, and IHPC, using their official train/validation/test splits throughout. In the ECG task, Lead-II signals are used as functional inputs to predict the QRS duration—a clinically informative measure of ventricular depolarization relevant for detecting conduction abnormalities (Hummel et al., 2009; Kashani & Barold, 2005). For the Tecator dataset, near-infrared absorbance spectra of meat samples serve as functional inputs to predict water content (Thodberg, 2015). In the Bike rental forecasting task (Fanaee-T, 2013), the daily rental-demand profile (a

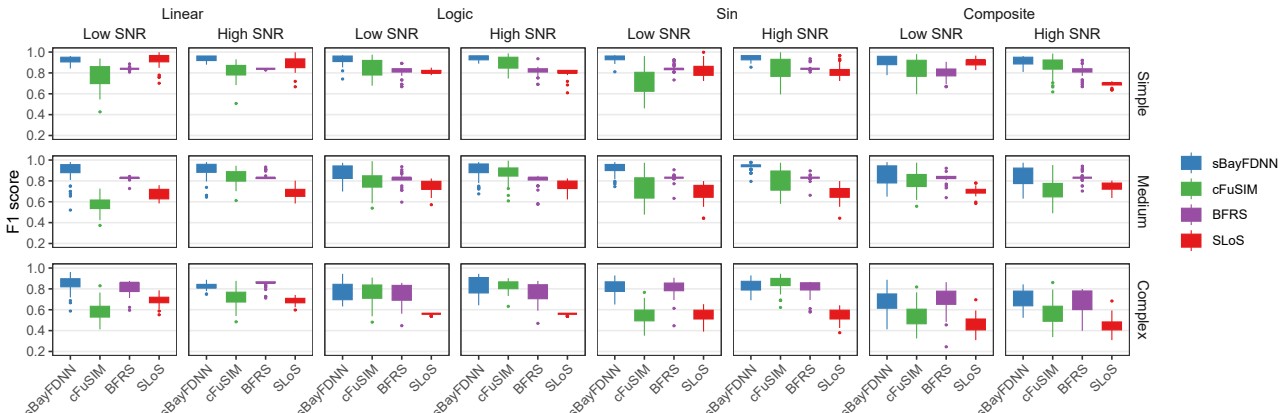

*Figure 2.* F1 scores across $g$ functions, SNR settings, and $\beta(t)$ scenarios.

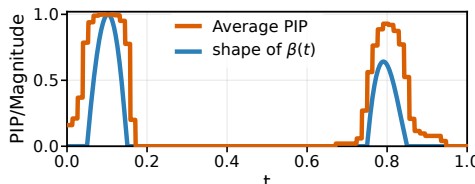

*Figure 3.* PIPs from sBayFDNN in the high-SNR, Medium-$\beta$, logistic-link scenario.

24-point curve) is used to predict total rentals over the following 7 days. For the IHPC dataset (Hebrail & Berard, 2006), daily minute-averaged active-power trajectories are taken as functional inputs to predict the next-day total energy usage. We report standard predictive metrics (RMSE/MAE) for all datasets. For ECG and Tecator, silver-standard region annotations are available, allowing us to also evaluate region-identification metrics. Detailed data information are provided in Appendix.

As shown in Table 1 and 2, sBayFDNN delivers the strongest overall performance across all datasets. While all methods attain relatively low precision in region selection, sBayFDNN achieves substantially higher recall and F1. Figure 5 plots the estimated PIPs on the original domains with sBayFDNN for ECG and Tecator. For ECG, sBayFDNN assigns higher inclusion strength within and near the clinically motivated QRS interval (shaded), with PIP values approaching 1 close to the boundaries. This is consistent with the fact that QRS duration is determined by onset/offset timing, making the endpoint morphology the most informative. Although the selected region is somewhat wider than the predefined silver interval, we do not interpret this as a localization failure. The QRS silver interval is a simplified fixed proxy, whereas the predictive morphology for QRS duration may extend into adjacent transition regions—particularly around onset/offset boundaries—which also carry clinically

relevant information. For Tecator, the PIP increases within the predefined water band (965–985 nm; shaded), while additional elevated PIP regions appear at earlier wavelengths (including ∼930 nm, often regarded as lipid/fat-associated in short-wave NIR), plausibly reflecting strong collinearity/compositional coupling between water and fat and other broad predictive structure. These results demonstrate that sBayFDNN can recover physically interpretable regions without sacrificing predictive accuracy.

## 7. Conclusion

We have presented a sparse Bayesian functional DNN framework for nonlinear scalar-on-function regression with automatic region selection. By integrating B-spline expansions with a Bayesian neural network and imposing a structured spike-and-slab prior, the proposed model captures complex nonlinear dependence between functional predictors and scalar responses through data-driven functional representations and a flexible deep architecture. The framework yields interpretable region-wise selection together with uncertainty quantification, supported by theoretical guarantees. Simulations and experiments on multiple real-world datasets confirm its selection accuracy and competitive predictive performance, demonstrating practical utility in identifying region-specific functional effects.

At the same time, several limitations should be noted. First, the method depends on the B-spline representation, including the choice of $J_n$, which controls a bias–resolution trade-off; although our additional analysis shows that the method does not collapse to pointwise screening at larger resolutions, fully adaptive basis selection remains an important direction for future work. Second, the current framework focuses on a single functional predictor and a single projection direction within each selected region. Extending it to multiple functional inputs, or to richer within-region embeddings with multiple projection directions, is natural but

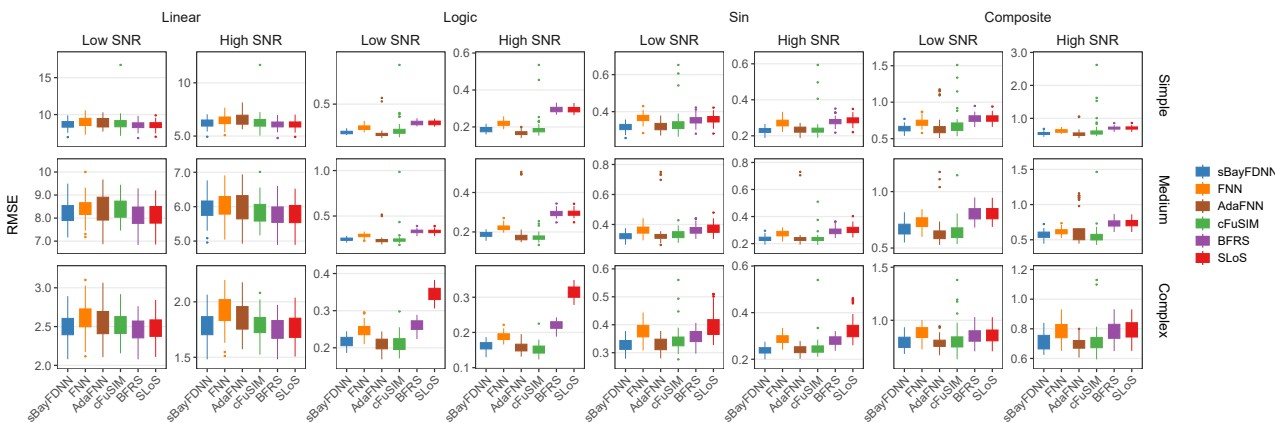

*Figure 4.* RMSE across $g$ functions, SNR settings, and $\beta(t)$ scenarios.

*Table 1.* Performance on ECG and Tecator datasets. F1, Recall, and Precision are reported only for methods that output an estimated active region; otherwise shown as "–". Best results within each dataset/metric are in bold.

| Method | ECG | | | | | Tecator | | | | |
|---|---|---|---|---|---|---|---|---|---|---|
| | RMSE | MAE | F1 | Recall | Precision | RMSE | MAE | F1 | Recall | Precision |
| sBayFDNN | **12.069** | **8.711** | **0.634** | **1.000** | **0.464** | **2.138** | **1.594** | **0.339** | **1.000** | **0.204** |
| FNN | 12.991 | 9.239 | – | – | – | 2.217 | 1.613 | – | – | – |
| AdaFNN | 14.083 | 10.198 | – | – | – | 3.027 | 2.299 | – | – | – |
| cFuSIM | 17.677 | 12.861 | 0.501 | 1.000 | 0.334 | 3.932 | 3.283 | 0.137 | 0.977 | 0.074 |
| BFRS | 16.297 | 11.805 | 0.396 | 0.784 | 0.265 | 2.691 | 2.250 | 0.211 | 0.714 | 0.124 |
| SLoS | 16.258 | 11.782 | 0.412 | 0.815 | 0.276 | 2.567 | 2.068 | 0.228 | 0.854 | 0.132 |

*Table 2.* Performance on Bike rental and IHPC datasets.

| Method | Bike | | IHPC | |
|---|---|---|---|---|
| | RMSE | MAE | RMSE | MAE |
| sBayFDNN | **0.618** | **0.497** | **0.536** | **0.409** |
| FNN | 0.699 | 0.535 | 0.549 | 0.409 |
| AdaFNN | 0.720 | 0.577 | 0.552 | 0.420 |
| cFuSIM | 0.693 | 0.539 | 0.548 | 0.411 |
| BFRS | 0.749 | 0.550 | 0.552 | 0.416 |
| SLoS | 0.684 | 0.506 | 0.549 | 0.411 |

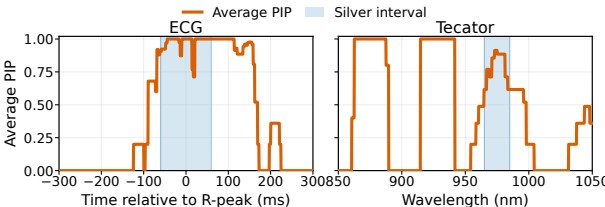

*Figure 5.* PIPs from sBayFDNN for ECG and Tecator datasets.

raises additional challenges in identifiability, since the relevant object may become a region-wise subspace rather than a single direction. Addressing this setting will likely require extra structural constraints such as orthogonality, group sparsity, or subspace-level regularization. Third, region-level interpretation in real applications depends on the covariance structure of the functional predictor; in settings with strong local collinearity, the selected regions are more appropriately viewed as data-supported predictive regions rather than oracle-exact support recovery, and should therefore be interpreted together with domain knowledge.

Several extensions are of practical interest. One is to jointly model functional and discrete covariates rather than pre-adjusting for potential confounders. Another is to replace the current $l_2$ loss with a more robust alternative for noisy or heavy-tailed settings. Finally, embedding the framework in generalized linear or survival-type models would broaden its applicability to binary, count, and time-to-event outcomes, further extending its utility in functional data analysis.

## Software and Data

The implementation code for sBayFDNN is available at https://github.com/mengyunwu2020/sBayFDNN, while the download link for the accompanying real dataset is provided in Appendix D.

## Acknowledgments

We thank the Area Chair and the anonymous reviewers for their insightful feedback, which was instrumental in strengthening this paper. This research was supported by the MOE Project of Humanities and Social Sciences (25YJCZH291); Shanghai Science and Technology Development Funds (23JC1402100); Shanghai Research Center for Data Science and Decision Technology; National Institutes of Health (CA204120); and National Science Foundation (2209685).

## Impact Statement

This work introduces a Bayesian deep learning framework for interpretable region selection in functional data. It advances nonlinear function-to-scalar regression by integrating structured sparsity with principled uncertainty quantification. The primary aim is to enhance machine learning methodology for functional and structured data. The proposed framework benefits domains where interpretability and reliability are essential, such as spectral chemometrics, neural signal analysis, and clinical monitoring. By enabling precise identification of informative functional subdomains, the method can support more accurate material composition analysis, refined physiological signal interpretation, and more targeted diagnostic interventions.

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

# A. Details for the Posterior Inference

## A.1. Hyperparameter Settings

All responses $Y$ are standardized using training-set statistics; unless otherwise stated, all computations are carried out on this standardized scale and we set the noise variance to $\sigma_\varepsilon^2 = 1$. For non-sparsified network parameters, we use independent Gaussian priors with variance $\sigma^2 = 1$ (numerically equal to $\sigma_\varepsilon^2$ under standardization, but conceptually distinct). We use $R = 5$ random restarts for each $J_n$.

**Simulation defaults.**

- Basis: B-splines with degree 4 (unless otherwise stated).

- Network: fully-connected ReLU, 64–64–64–1; first-layer column-wise spike-and-slab selection.

- Optimizer: mini-batch SGD, learning rate $10^{-3}$, batch size 64, max 80,001 iterations, early stopping patience 3,000.

- Spike-and-slab: $(\lambda_n, \sigma_{0,n}^2, \sigma_{1,n}^2) = (10^{-5}, 10^{-5}, 2 \times 10^{-3})$.

We treat the projection truncation level $J_n$ as a resolution hyperparameter and select it via restart-aggregated evidence in simulations or restart-aggregated validation loss when evidence computation is prohibitive. We use the candidate set $\mathcal{J} = \{55, 60, 70, 80\}$ for $J_n$ in simulations; for real datasets, $\mathcal{J}$ is chosen as a small neighborhood around a dataset-specific baseline resolution. This choice is further supported by an additional exploration of $J_n$ sensitivity across replications in Appendix A.4.

We train with a small learning rate and a relatively large early-stopping patience to obtain stable solutions across restarts. The sparsification prior is intentionally strong in simulations; empirically, the same default sparsification hyperparameters remain effective on ECG ($n \approx 2\text{–}3 \times 10^4$), consistent with increasingly data-dominated inference as $n$ grows.

For competing methods, we aim to ensure fair comparisons by using comparable model capacity whenever applicable (e.g., similar depth/width for neural-network baselines). Method-specific hyperparameters are taken from authors' recommended defaults when available, and otherwise selected by cross-validation or validation loss on the same training/validation split.

## A.2. Stochastic Gradient Algorithm (SGD) for the MAP Fitting

We first provide the derivations on the marginal prior on $\boldsymbol{\theta}$. Denote $K_{1_n} = (J_n + 1)L_1 + \sum_{k=2}^{H_n}(L_{k-1} + 1)L_k$ as the dimension of $\boldsymbol{\theta}$ and $\mathcal{W} \subset \{1, \ldots, K_{1_n}\}$ as the set of indices in $\boldsymbol{\theta}$ corresponding to the weights of the first-layer $\{W_{1,hg}\}_{h \leq L_1, g \leq J_n}$, and let $\mathcal{G} := \{1, \ldots, K_{1_n}\} \setminus \mathcal{W}$. In addition, let $\phi(\cdot; 0, s^2)$ denote the $\mathcal{N}(0, s^2)$ density.

Based on the priors introduced in Section 3, we have

$$\pi(\boldsymbol{W}_{1,*j} \mid \gamma_j) = \prod_{h=1}^{L_1} \left[\phi(W_{1,hj}; 0, \sigma_{1,n}^2)\right]^{\gamma_j} \left[\phi(W_{1,hj}; 0, \sigma_{0,n}^2)\right]^{1-\gamma_j}, \forall j \in \{1, \ldots, J_n\};$$

$$\pi(\gamma_j) = \lambda_n^{\gamma_j}(1 - \lambda_n)^{1-\gamma_j};$$

$$\pi(W_{h,ab}) = \phi(W_{h,ab}; 0, \sigma^2), \qquad \forall h \in \{2, \ldots, H_n\};$$

and

$$\pi(b_{h,a}) = \phi(b_{h,a}; 0, \sigma^2), \qquad \forall h \in \{1, \ldots, H_n\}.$$

Then, the marginal prior on $\boldsymbol{\theta}$ is $\pi(\boldsymbol{\theta}) = \sum_{\boldsymbol{\gamma}} \pi(\boldsymbol{\theta}, \boldsymbol{\gamma})$, where

$$\pi(\boldsymbol{\theta}, \boldsymbol{\gamma}) = \prod_{j=1}^{J_n} [\pi(\gamma_j)\pi(\boldsymbol{W}_{1,*j} \mid \gamma_j)] \prod_{j \in \mathcal{G}} \phi(\theta_j; 0, \sigma^2).$$

Based on the marginal prior on $\boldsymbol{\theta}$, we develop the following SGD algorithm (Algorithm 1) for optimizing (9).

---

**Algorithm 1** Sparse DNN elicitation with projection-size selection

---

**Input:** $\mathcal{D}_{\mathrm{tr}} = \{(X_i, Y_i)\}_{i=1}^{n_{\mathrm{tr}}}$, $\mathcal{D}_{\mathrm{va}} = \{(X_i, Y_i)\}_{i=1}^{n_{\mathrm{va}}}$; candidate projection dimensions $\mathcal{J} = \{J_1, \ldots, J_M\}$; basis specification for $\{B_j(t)\}_{j \geq 1}$; random restarts $R$; hyperparameters $\sigma_{0,n}^2, \sigma_{1,n}^2, \sigma^2, \lambda_n$; noise variance $\sigma_\varepsilon^2$; Crit $\in \{\text{evidence}, \text{val}\}$.

**Output:** Selected projection dimension $J^\star$, aggregated predictor $\widehat{f}(\cdot)$, averaged inclusion scores $\widehat{\mathbf{q}}$, and selected feature mask $\widehat{\boldsymbol{\gamma}}$.

---

**foreach** $J \in \mathcal{J}$ **do**

  **Step 0 (projection).** Construct $\{B_1(t), \ldots, B_J(t)\}$ and compute $\boldsymbol{\eta}_J(X)$ for all $(X, Y) \in \mathcal{D}_{\mathrm{tr}} \cup \mathcal{D}_{\mathrm{va}}$.

  **for** $r = 1, \ldots, R$ **do**

    **Step 1 (initialize).** Randomly initialize $\boldsymbol{\theta}$.

    **Step 2 (MAP training).** Obtain

$$\widehat{\boldsymbol{\theta}}_{J,r} \in \arg\min_{\boldsymbol{\theta}} \ \mathcal{L}_{n,J}(\boldsymbol{\theta}) = \frac{1}{2\sigma_\varepsilon^2} \sum_{(X_i, Y_i) \in \mathcal{D}_{\mathrm{tr}}} \Big(Y_i - F_{\boldsymbol{\theta}}(\boldsymbol{\eta}_J(X_i))\Big)^2 - \log \pi(\boldsymbol{\theta}).$$

    **Step 3 (validation score).** Set $v_{J,r} := \mathrm{MSE}\Big(\widehat{\boldsymbol{\theta}}_{J,r}; \mathcal{D}_{\mathrm{va}}\Big)$.

    **Step 4 (feature mask and evidence surrogate).** Let $\widehat{\mathbf{W}}_{J,r}^{(1)} \in \mathbb{R}^{w \times J}$ be the first-layer weight matrix in $\widehat{\boldsymbol{\theta}}_{J,r}$. For $j = 1, \ldots, J$, set

$$\gamma_{J,r,j} = \mathbb{1}\Big(\big\|\widehat{\mathbf{W}}_{J,r,:,j}^{(1)}\big\|_2^2 > \tau_n\Big), \quad \tau_n := \frac{\log\big((1-\lambda_n)/\lambda_n\big) + \frac{w}{2}\log(\sigma_{1,n}^2/\sigma_{0,n}^2)}{\frac{1}{2\sigma_{0,n}^2} - \frac{1}{2\sigma_{1,n}^2}}.$$

    Denote $\widehat{\boldsymbol{\gamma}}_{J,r} := (\gamma_{J,r,1}, \ldots, \gamma_{J,r,J})^\top$ and construct the evidence surrogate $\widehat{\boldsymbol{\theta}}_{J,r}^{\mathrm{s}}$ by replacing $\widehat{\mathbf{W}}_{J,r}^{(1)}$ with $\widehat{\mathbf{W}}_{J,r}^{(1)}\mathrm{diag}(\widehat{\boldsymbol{\gamma}}_{J,r})$ (used only for evidence computation).

    **Step 5 (evidence score; post-sparsification).** Compute $\ell_{J,r} := \log \mathrm{Ev}\Big(\widehat{\boldsymbol{\theta}}_{J,r}^{\mathrm{s}}; \mathcal{D}_{\mathrm{tr}}\Big)$.

  **Step 6 (aggregate over restarts).** Set $\bar{\ell}_J := R^{-1} \sum_{r=1}^R \ell_{J,r}$ and $\bar{v}_J := R^{-1} \sum_{r=1}^R v_{J,r}$.

**Step 7 (select $J$).** **if** Crit $=$ *evidence* **then**

  $J^\star \in \arg\max_{J \in \mathcal{J}} \ \bar{\ell}_J$.

**else**

  $J^\star \in \arg\min_{J \in \mathcal{J}} \ \bar{v}_J$.

**Step 8 (aggregate predictor and PIPs at $J^\star$).** Define the aggregated predictor $\widehat{f}(x) := \frac{1}{R} \sum_{r=1}^R F_{\widehat{\boldsymbol{\theta}}_{J^\star,r}}(\boldsymbol{\eta}_{J^\star}(x))$.

For each $r = 1, \ldots, R$ and $j = 1, \ldots, J^\star$, compute the restart-specific plug-in inclusion score $q_{J^\star, r, j}$ from $\widehat{\mathbf{W}}_{J^\star, r, :, j}^{(1)}$.

Compute the restart-specific plug-in inclusion score $q_{J^\star, r, j}$ then set $\widehat{q}_j := \frac{1}{R} \sum_{r=1}^R q_{J^\star, r, j}, j = 1, \ldots, J^\star$.

Let $\widehat{\mathbf{q}} := (\widehat{q}_1, \ldots, \widehat{q}_{J^\star})^\top$ and define the final feature mask $\widehat{\gamma}_j = \mathbb{1}(\widehat{q}_j > \tau), j = 1, \ldots, J^\star$, for a prespecified thresholding rule.

**return** $(J^\star, \widehat{f}, \widehat{\mathbf{q}}, \widehat{\boldsymbol{\gamma}})$.

---

After selecting $J^\star$, we form the final predictor by averaging the outputs of the $R$ independently trained models at $J^\star$, i.e., $\hat{y}(x) = \frac{1}{R}\sum_{r=1}^{R} F_{\widehat{\boldsymbol{\theta}}_{J^\star,r}}(\boldsymbol{\eta}_{J^\star}(x))$. For interpretability summaries, we first compute restart-specific plug-in PIPs at the selected resolution $J^\star$, average them across the same $R$ fitted models, and then map the averaged PIPs to the functional domain for region-level interpretation.

In Step 5 of Algorithm 1, we score each run using a Laplace-type evidence surrogate (MacKay, 1992; Liang et al., 2013). This score is used only as a practical model-comparison criterion across candidate projection dimensions and random restarts, rather than as an exact marginal likelihood for the full nonconvex posterior.

Let $\mathcal{L}_{n,J}(\boldsymbol{\theta})$ denote the negative log-posterior objective used in MAP training under projection dimension $J$, and define $h_{n,J}(\boldsymbol{\theta}) := -\mathcal{L}_{n,J}(\boldsymbol{\theta})/n_{\text{tr}}$. Let $\boldsymbol{H}_{n,J}(\boldsymbol{\theta}) := \nabla^2_{\boldsymbol{\theta}} h_{n,J}(\boldsymbol{\theta})$. For this score, we evaluate $h_{n,J}$ and $\boldsymbol{H}_{n,J}$ at the sparsified surrogate parameter $\widehat{\boldsymbol{\theta}}^{\text{s}}_{J,r}$ constructed in Step 4, while predictive evaluations are based on the original MAP estimate $\widehat{\boldsymbol{\theta}}_{J,r}$.

Let $\mathcal{I}_{J,r}$ index the parameters retained in the sparsified surrogate, and let $d_{J,r} := |\mathcal{I}_{J,r}|$. Denote by $\boldsymbol{H}_{n,J}(\boldsymbol{\theta})_{\mathcal{I}_{J,r},\mathcal{I}_{J,r}}$ the corresponding principal submatrix. The Laplace-type log-score is defined as

$$\ell_{J,r} := n_{\text{tr}} h_{n,J}\left(\widehat{\boldsymbol{\theta}}^{\text{s}}_{J,r}\right) + \frac{d_{J,r}}{2}\log(2\pi) - \frac{d_{J,r}}{2}\log(n_{\text{tr}}) - \frac{1}{2}\log\det\left(-\boldsymbol{H}_{n,J}\left(\widehat{\boldsymbol{\theta}}^{\text{s}}_{J,r}\right)_{\mathcal{I}_{J,r},\mathcal{I}_{J,r}}\right). \tag{10}$$

Here the Hessian is restricted to the retained parameters of the sparsified surrogate. Because the underlying DNN posterior is nonconvex and potentially multimodal, $\ell_{J,r}$ should be interpreted as a computationally tractable Laplace-type selection score, not as exact Bayesian evidence.

We compute the log-determinant term in (10) using the eigenvalues of the restricted negative Hessian for numerical stability, and apply standard stabilization when needed, such as adding a small diagonal jitter.

## A.3. Examination of Evidence-Based $J_n$ Selection

To assess the practical effectiveness of the evidence-based criterion for selecting $J_n$, we examine how highly the selected candidate ranks relative to the oracle choice defined by F1 over the candidate set. Based on 50 replicates, Table 3 summarizes the resulting seed-level rankings across three representative scenarios with composite $g$ and SNR$= 10$ (see Sections 6.1 and C.2 for details). Since the candidate set contains eight possible $J_n$ values in each run, this analysis is intended to evaluate whether the evidence-based criterion tends to identify competitive choices, rather than to claim exact recovery of the oracle $J_n$ in every case.

Table 3. Top-$k$ effectiveness of evidence-based $J_n$ selection (seed-level; F1 criterion). Candidate set size per run is 8.

| Scenario | Mean Rank | Top-1 | Top-2 | Top-3 | Within 15% of Oracle F1 | Within 30% of Oracle F1 |
|---|---|---|---|---|---|---|
| Complex $\beta(t)$, Composite $g$ | 3.85 | 11.0% | 21.4% | 28.6% | 50.0% | 92.9% |
| Simple $\beta(t)$, Composite $g$ | 2.5 | 28.6% | 42.9% | 64.3% | 100.0% | 100.0% |
| Moderate $\beta(t)$, Composite $g$ | 3.33 | 33.3% | 33.3% | 33.3% | 66.7% | 100.0% |
| Overall | 3.17 | 17.6% | 23.5% | 44.1% | 73.5% | 97.1% |

The results suggest that the evidence-based criterion is reasonably effective in practice. Although it does not always identify the oracle-ranked $J_n$, the selected candidate is often competitive: overall, 44.1% of the selections fall within the top three candidates, 73.5% are within 15% of the oracle F1, and 97.1% are within 30% of the oracle F1. This indicates that the criterion frequently yields near-oracle choices, especially in the simpler scenarios, while leaving room for future exploration of potentially stronger model-selection rules.

## A.4. Sensitivity Analysis of $J_n$

To examine how the number of spline bases $J_n$ affects support recovery and prediction performance, we perform a sensitivity analysis under two simulation scenarios: simple $\beta(t)$ with composite $g$ at SNR $= 10$ and SNR $= 5$. For each scenario, we generated 50 independent data replications and evaluated $J_n \in \{20, 40, \ldots, 240\}$. For each replication and each $J_n$, we fit the model once and computed region-selection and prediction metrics. We then summarized results as mean(sd) over the 50 replications.

In addition to the standard recovery metrics, we also consider two measures to characterize the continuity and granularity of the resulting importance profile and selected regions. Specifically, *Curve Roughness* quantifies the roughness of

the importance curve $p(t)$ through $\frac{1}{T-1}\sum_{k=1}^{T-1}\left|p(t_{k+1}) - p(t_k)\right|$, where larger values indicate a less smooth and more fragmented importance curve. We also report *MinLen/Mesh*, defined as the minimum selected interval length divided by the mesh size. This measures how many grid cells are covered by the shortest selected interval, with smaller values indicating more fragmented selections. Together, these two metrics help assess whether increasing $J_n$ leads to more irregular importance profiles or excessively short selected intervals. A third metric, *MeanLen* (the mean interval length of the identified region), is also reported.

*Table 4.* Simulation results under different $J_n$ settings for the scenario with simple $\beta(t)$, composite link function $g$, and two SNR levels (reported as mean(sd) over 50 reps).

| SNR | $J_n$ | MeanLen | Curve Roughness | RMSE | F1 | Recall | Precision | MinLen/Mesh |
|---|---|---|---|---|---|---|---|---|
| | 20 | 0.5500(0.1843) | 0.0056(0.0014) | 0.5587(0.0640) | 0.5573(0.1017) | 1.0000(0.0000) | 0.3926(0.0924) | 11.0000(3.6857) |
| | 40 | 0.2800(0.0307) | 0.0077(0.0005) | 0.5515(0.0518) | 0.8366(0.0523) | 1.0000(0.0000) | 0.7225(0.0780) | 11.2000(1.2262) |
| | 60 | 0.2286(0.0300) | 0.0077(0.0005) | 0.5522(0.0493) | 0.9175(0.0437) | 0.9807(0.0456) | 0.8684(0.0823) | 13.7143(1.7981) |
| | 80 | 0.2087(0.0337) | 0.0077(0.0012) | 0.5677(0.0496) | 0.9323(0.0480) | 0.9500(0.0568) | 0.9245(0.0921) | 16.6947(2.6985) |
| | 100 | 0.1782(0.0352) | 0.0084(0.0015) | 0.5708(0.0513) | 0.9095(0.0477) | 0.8867(0.0875) | 0.9462(0.0783) | 17.1875(4.9006) |
| 10 | 120 | 0.1489(0.0402) | 0.0083(0.0014) | 0.5698(0.0535) | 0.8803(0.0576) | 0.8229(0.0997) | 0.9601(0.0652) | 16.4276(6.9745) |
| | 140 | 0.1360(0.0366) | 0.0097(0.0025) | 0.5647(0.0486) | 0.8720(0.0611) | 0.7928(0.0965) | 0.9797(0.0459) | 17.0265(7.9199) |
| | 160 | 0.1084(0.0405) | 0.0102(0.0026) | 0.5733(0.0540) | 0.8477(0.0719) | 0.7505(0.1032) | 0.9854(0.0384) | 14.4821(8.9812) |
| | 180 | 0.0921(0.0399) | 0.0106(0.0026) | 0.5764(0.0552) | 0.8180(0.0685) | 0.7014(0.0963) | 0.9927(0.0318) | 13.4591(9.4446) |
| | 200 | 0.0707(0.0298) | 0.0112(0.0025) | 0.5751(0.0558) | 0.8021(0.0714) | 0.6783(0.0992) | 0.9946(0.0227) | 9.7347(7.5132) |
| | 220 | 0.0605(0.0278) | 0.0114(0.0021) | 0.5689(0.0532) | 0.7702(0.0618) | 0.6325(0.0806) | 0.9949(0.0192) | 9.2074(7.3905) |
| | 240 | 0.0569(0.0289) | 0.0127(0.0029) | 0.5724(0.0533) | 0.7583(0.0687) | 0.6210(0.0889) | 0.9862(0.0373) | 9.2339(7.9701) |
| | 20 | 0.5475(0.1659) | 0.0058(0.0014) | 0.6542(0.0661) | 0.5549(0.0928) | 1.0000(0.0000) | 0.3892(0.0841) | 10.9500(3.3174) |
| | 40 | 0.2878(0.0336) | 0.0077(0.0004) | 0.6474(0.0515) | 0.8237(0.0543) | 1.0000(0.0000) | 0.7037(0.0777) | 11.5111(1.3424) |
| | 60 | 0.2354(0.0376) | 0.0075(0.0006) | 0.6457(0.0491) | 0.9090(0.0601) | 0.9843(0.0343) | 0.8528(0.1059) | 14.1214(2.2567) |
| | 80 | 0.2129(0.0364) | 0.0077(0.0008) | 0.6587(0.0510) | 0.9240(0.0495) | 0.9571(0.0591) | 0.9032(0.0976) | 16.9263(3.2311) |
| | 100 | 0.1732(0.0351) | 0.0084(0.0016) | 0.6611(0.0513) | 0.9037(0.0551) | 0.8802(0.0927) | 0.9414(0.0819) | 16.5417(5.0492) |
| 5 | 120 | 0.1468(0.0437) | 0.0088(0.0020) | 0.6646(0.0542) | 0.8808(0.0546) | 0.8262(0.0962) | 0.9549(0.0621) | 16.0138(7.4076) |
| | 140 | 0.1299(0.0415) | 0.0098(0.0025) | 0.6602(0.0537) | 0.8693(0.0601) | 0.7904(0.0919) | 0.9750(0.0473) | 16.1000(8.2870) |
| | 160 | 0.0983(0.0381) | 0.0100(0.0026) | 0.6613(0.0500) | 0.8355(0.0667) | 0.7374(0.1006) | 0.9767(0.0522) | 12.7179(8.2988) |
| | 180 | 0.0889(0.0396) | 0.0105(0.0024) | 0.6663(0.0552) | 0.8169(0.0748) | 0.7009(0.1046) | 0.9916(0.0346) | 12.7432(9.2985) |
| | 200 | 0.0721(0.0350) | 0.0115(0.0021) | 0.6685(0.0554) | 0.7980(0.0755) | 0.6756(0.1006) | 0.9877(0.0362) | 10.4286(8.6346) |
| | 220 | 0.0663(0.0325) | 0.0115(0.0026) | 0.6614(0.0512) | 0.7699(0.0685) | 0.6356(0.0920) | 0.9901(0.0311) | 10.5926(8.8314) |
| | 240 | 0.0524(0.0280) | 0.0128(0.0028) | 0.6633(0.0550) | 0.7497(0.0677) | 0.6095(0.0863) | 0.9870(0.0358) | 9.0305(7.6008) |

Table 4 shows a consistent bias–resolution trade-off. When $J_n$ is small, selected regions are relatively broad (larger mean interval length), which tends to increase recall but reduce precision. As $J_n$ increases to a moderate range, localization improves and F1 reaches its best levels. For very large $J_n$, selected regions become increasingly fragmented (smaller interval length and lower MinLen/Mesh), and recall gradually decreases, leading to lower F1. Importantly, even at larger $J_n$, selected regions are still induced by overlapping spline supports in the function domain, rather than pointwise discrete screening. Overall, these results support the practical criterion used in the main text: moderate $J_n$ values provide a better balance between interpretability (region continuity) and selection accuracy.

### A.5. Sensitivity Analysis of Hyperparameters

We conduct one-at-a-time (OAT) sensitivity analyses by varying one hyperparameter at a time while fixing the others at their default values. Specifically, simulated data under various settings (see Sections 6.1 and C.2 for details) are examined with 50 replications per scenario. Summarized results for the spike-and-slab hyperparameters ($\lambda_n$, $\sigma_0^2$, and $\sigma_1^2$) and the PIP threshold ($\tau$) are provided in Tables 5 and 6, respectively, and the results in both tables are macro-averaged over all simulation scenarios.

For the spike-and-slab hyperparameters, the prediction and recovery results remain reasonably stable across the examined ranges, suggesting that the proposed method is not overly sensitive to moderate changes in these prior hyperparameters. For $\lambda_n$, no clear monotone trend is observed, and the macro-averaged RMSE and recovery metrics remain broadly comparable across values, indicating relative robustness to the prior inclusion probability. In contrast, $\sigma_0^2$ exhibits a clearer effect on the recall–precision trade-off: an extremely small spike variance leads to overly inclusive selection, with near-perfect recall but much lower precision, whereas larger values make the selection more conservative. For $\sigma_1^2$, the effect is also

non-monotone, but different values induce different recall–precision trade-offs; the default choice $\sigma_1^2 = 2 \times 10^{-3}$ provides the most balanced overall performance in terms of macro-averaged F1.

As expected, $\tau$ controls the trade-off between recall and precision (Table 6). When $\tau$ is small (e.g., $\tau = 0.1$), the procedure is more inclusive, leading to very high recall but lower precision and a larger number of selected components. As $\tau$ increases to a moderate range ($\tau = 0.2$–$0.5$), the method achieves a more balanced recovery performance, with the best F1 scores attained at $\tau = 0.2$ and $\tau = 0.3$. When $\tau$ becomes too large ($\tau \geq 0.6$), the selection becomes increasingly conservative, resulting in substantially reduced recall and fewer selected components, which in turn leads to a marked drop in F1. Overall, these results suggest that the method is reasonably stable over a moderate range of threshold values, while overly small or overly large thresholds may induce over-selection or under-selection, respectively.

*Table 5.* Sensitivity analysis of the spike-and-slab hyperparameters (macro-averaged over scenarios). Results are reported as mean(sd).

| Hyperparameter | Value | RMSE | F1 | Recall | Precision |
|---|---|---|---|---|---|
| $\lambda_n$ | $1 \times 10^{-6}$ | 0.6747(0.1180) | 0.7868(0.1472) | 0.7574(0.1582) | 0.8349(0.1261) |
| | $3 \times 10^{-6}$ | 0.6813(0.1273) | 0.8182(0.2748) | 0.7859(0.2667) | 0.8532(0.1764) |
| | $1 \times 10^{-5}$ | 0.6693(0.1201) | 0.7982(0.1071) | 0.8884(0.1045) | 0.7346(0.1007) |
| | $3 \times 10^{-5}$ | 0.6931(0.1237) | 0.7165(0.1995) | 0.6845(0.2687) | 0.8470(0.0874) |
| | $1 \times 10^{-4}$ | 0.6743(0.1183) | 0.7947(0.1382) | 0.7746(0.1438) | 0.8315(0.1228) |
| $\sigma_0^2$ | $1 \times 10^{-6}$ | 0.6542(0.1271) | 0.5463(0.0754) | 0.9990(0.0010) | 0.3826(0.0732) |
| | $3 \times 10^{-6}$ | 0.6889(0.1341) | 0.7142(0.2248) | 0.6796(0.3045) | 0.8437(0.0551) |
| | $1 \times 10^{-5}$ | 0.6693(0.1201) | 0.7982(0.1071) | 0.8884(0.1045) | 0.7346(0.1007) |
| | $3 \times 10^{-5}$ | 0.6937(0.1229) | 0.6684(0.2178) | 0.5682(0.2576) | 0.8952(0.0752) |
| | $1 \times 10^{-4}$ | 0.6873(0.1232) | 0.6991(0.1973) | 0.6642(0.2592) | 0.7380(0.0322) |
| $\sigma_1^2$ | $1 \times 10^{-3}$ | 0.6920(0.1195) | 0.7035(0.1536) | 0.6346(0.2215) | 0.7892(0.0238) |
| | $1.5 \times 10^{-3}$ | 0.6832(0.1256) | 0.7150(0.2056) | 0.6785(0.3121) | 0.8606(0.0016) |
| | $2 \times 10^{-3}$ | 0.6693(0.1201) | 0.7982(0.1071) | 0.8884(0.1045) | 0.7346(0.1007) |
| | $3 \times 10^{-3}$ | 0.6928(0.1303) | 0.7095(0.2294) | 0.6742(0.3097) | 0.8470(0.0529) |
| | $5 \times 10^{-3}$ | 0.6534(0.1278) | 0.7759(0.1444) | 0.9854(0.0028) | 0.6696(0.2009) |

*Table 6.* Sensitivity analysis of the PIP threshold $\tau$ (macro-averaged over scenarios). Results are reported as mean(sd).

| $\tau$ | F1 | Recall | Precision | $n_{\text{selected}}$ |
|---|---|---|---|---|
| 0.1 | 0.8256(0.0217) | 0.9930(0.0031) | 0.7251(0.0316) | 11.51(0.48) |
| 0.2 | 0.8969(0.0137) | 0.9573(0.0083) | 0.8579(0.0244) | 8.30(0.21) |
| 0.3 | 0.8969(0.0137) | 0.9573(0.0083) | 0.8579(0.0244) | 8.30(0.21) |
| 0.4 | 0.8781(0.0126) | 0.8610(0.0164) | 0.9146(0.0201) | 5.83(0.19) |
| 0.5 | 0.8781(0.0126) | 0.8610(0.0164) | 0.9146(0.0201) | 5.83(0.19) |
| 0.6 | 0.7428(0.0281) | 0.6498(0.0340) | 0.9201(0.0159) | 3.32(0.22) |
| 0.7 | 0.7428(0.0281) | 0.6498(0.0340) | 0.9201(0.0159) | 3.32(0.22) |
| 0.8 | 0.4557(0.0407) | 0.3523(0.0353) | 0.6864(0.0497) | 1.20(0.13) |
| 0.9 | 0.4557(0.0407) | 0.3523(0.0353) | 0.6864(0.0497) | 1.20(0.13) |

## A.6. Computational Time Analysis of sBayFDNN on Simulated Data

Tables 7 and 8 report the computational cost of the proposed method for varying numbers of observed points $T_{obs}$ and sample size $n$, with the running time of sBayFDNN decomposed into projection, training, and evidence evaluation. It is observed that with a fixed sample size, increasing $T_{obs}$ from 200 to 2000 has little effect on the overall runtime. In particular, the projection step is negligible throughout, and the total runtime is dominated by training and evidence computation. This suggests that, once the functional input has been projected onto a fixed basis representation, the computational burden depends much more on the downstream optimization procedure than on the raw observation grid size itself.

By contrast, the sample size $n$ has a more visible impact on runtime. At fixed $T_{obs} = 200$, increasing $n$ from 1000 to 3000 leads to a clear increase in both training and evidence computation time, and hence in the overall runtime per repeat. This is consistent with the fact that the optimization and model-evaluation steps scale primarily with the number of samples rather than with the original number of observed time points.

*Table 7.* Runtime vs. observed points $T_{obs}$ at fixed $n = 1000$. Entries are mean(sd) over 5 repeats (seconds).

| $T_{obs}$ | Projection ($\times 10^{-3}$ s) | Training | Evidence | Runtime/Repeat |
|---|---|---|---|---|
| 200 | 2.0 (5.0) | 83.8 (22.1) | 41.3 (3.6) | 125.1 (22.8) |
| 400 | 2.0 (4.0) | 80.3 (24.8) | 41.5 (7.6) | 121.7 (20.1) |
| 600 | 2.0 (4.0) | 105.9 (66.0) | 38.8 (7.0) | 144.7 (62.8) |
| 1000 | 3.0 (6.0) | 68.4 (18.0) | 36.3 (3.7) | 104.7 (15.1) |
| 1500 | 2.0 (4.0) | 68.0 (18.5) | 36.2 (3.0) | 104.2 (15.8) |
| 2000 | 2.0 (5.0) | 67.6 (19.3) | 36.1 (3.1) | 103.7 (16.7) |

*Table 8.* Runtime vs. sample size $n$ at fixed $T_{obs} = 200$. Entries are mean(sd) over 5 repeats (seconds).

| $n$ | Projection ($\times 10^{-3}$ s) | Train | Evidence | Runtime/Repeat |
|---|---|---|---|---|
| 1000 | 2.0 (5.0) | 83.8 (22.1) | 41.3 (3.6) | 125.1 (22.8) |
| 2000 | 2.0 (5.0) | 122.2 (83.3) | 80.3 (56.9) | 202.5 (58.3) |
| 3000 | 3.0 (6.0) | 151.2 (97.5) | 49.4 (11.5) | 200.6 (96.3) |

## A.7. Cross-seed Stability of Plug-in PIPs and Region Selection

To assess whether plug-in PIPs and the resulting selected regions are sensitive to random initialization and optimization path, we perform a cross-seed stability analysis under the simple $\beta(t)$ and composite $g$ setting at SNR = 10 and SNR = 5. For each scenario, we generate 50 independent data replications, and within each replication we refit the model using 10 random seeds while keeping the dataset and all other settings fixed.

For each replication, we summarize cross-seed variability under two reference schemes: comparison to the seed-0 fit and comparison to the cross-seed mean. We report both PIP-level and selection-level stability metrics. Specifically, *PIP Corr* and *PIP MeanDiff* compare the estimated PIP curves across seeds, measuring similarity in shape and average numerical discrepancy, respectively. After thresholding the PIP curves to obtain selected regions, we further compute *Interval SymDiff*, namely the symmetric-difference length between two selected supports, and $\Delta$F1, which measures the seed-induced change in support recovery performance. These replication-level summaries are then aggregated over the 50 replications, and the resulting distributions are shown in Figure 6.

Figure 6 shows that cross-seed variability is low under both low and high SNR scenarios. In particular, the PIP correlations are consistently high, while the mean PIP differences and interval symmetric differences remain small. Moreover, $\Delta$F1 stays close to zero across seeds, indicating that the final region-selection performance is only mildly affected by initialization and optimization path. Overall, the plug-in PIP profiles and the resulting selected regions remain highly stable across seeds, with only slightly larger variability under low SNR scenario. These results provide empirical support that the MAP-based plug-in PIPs and the resulting selected regions are stable in practice.

## B. Proof of Statistical Properties

We divide the theoretical proof into four parts. First, we establish the theoretical foundation for finite-dimensional approximations of functional data, which lays the groundwork for the subsequent theorems. Then, in the second, third, and fourth parts, we provide proofs for the bound of the approximation error (Theorem 5.4), the posterior consistency (Theorem 5.7), and the selection consistency (Theorem 5.9), respectively. Prior to proving each theorem, we present necessary lemmas. Throughout, $C, C', C_1, \ldots$ denote generic positive constants whose values may change from line to line.

### B.1. Finite-dimensional Approximations

First, recall and introduce some notations. Let $\beta^*$ and $\beta^*_{J_n} = \sum_{j=1}^{J_n} \omega^*_{J_n,j} B_j(t)$ denote the true coefficient function and its truncated counterpart, respectively, where $\boldsymbol{\omega}^*_{J_n} = (\omega^*_{J_n,1}, \cdots, \omega^*_{J_n,J_n})^T$ is the vector of true basis coefficients for $\beta^*_{J_n}$. Denote $S^*_{J_n} = \text{supp}(\boldsymbol{\omega}^*_{J_n})$ and $\Omega^*_{J_n} = \cup_{j \in S^*_{J_n}} I_j$ with $I_j := \text{supp}(B_j)$. In addition, let $u^*(X) := \langle X, \beta^* \rangle$, $\mu^*(X) :=$ $g^*(u^*(X)), u^*_{J_n}(X) := \langle X, \beta^*_{J_n} \rangle$, and $\mu^*_{J_n}(X) := g^*(u^*_{J_n}(X))$. Moreover, for $\kappa > 0$, define the strong-signal region $\Omega^*(\kappa) := \{t \in [0,1] : |\beta^*(t)| > \kappa\}$. Define $S^*_{J_n}(\kappa) := \{j \in [J_n] : I_j \cap \Omega^*(\kappa) \neq \varnothing\}$ and $\Omega^*_{J_n}(\kappa) := \bigcup_{j \in S^*_{J_n}(\kappa)} I_j$. Let $\Delta_{J_n} := \max_{1 \le j \le J_n} \text{diam}(I_j) \asymp J_n^{-1}$ for fixed spline order. For a measurable set $A \subset [0,1]$ and $\Delta > 0$, define its $\Delta$-enlargement by $A^{+\Delta} := \{t \in [0,1] : \text{dist}(t, A) \le \Delta\}$. For the strong-signal cover $\Omega^*_{J_n}(\kappa)$, we will also use the

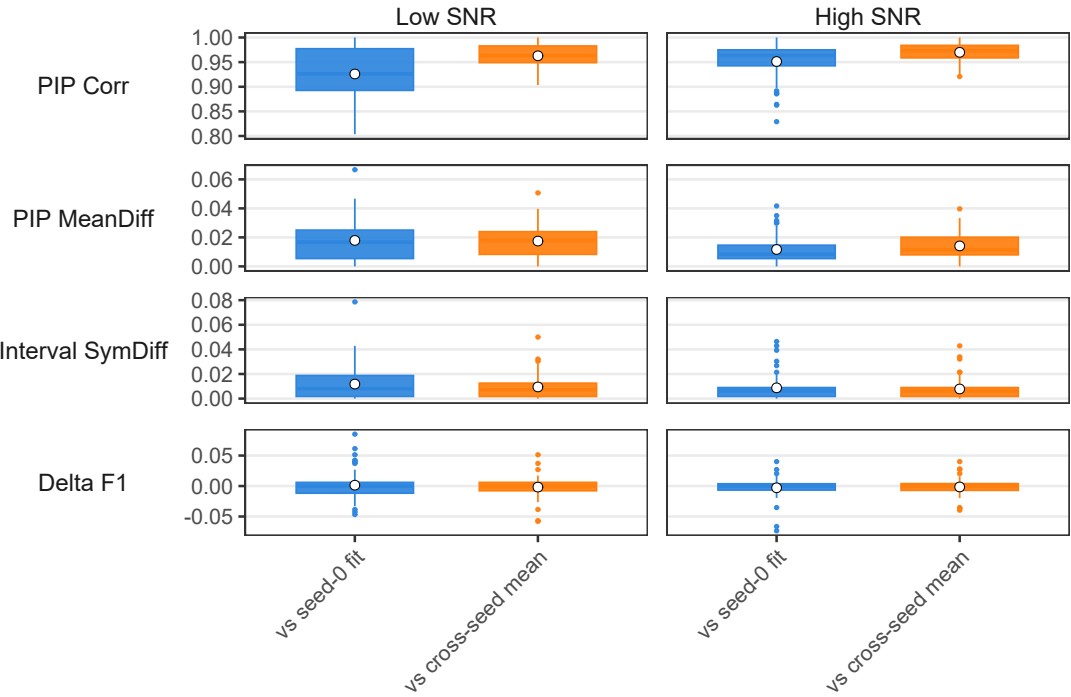

*Figure 6.* Cross-seed stability of the plug-in PIPs and the resulting selected regions under the simple $\beta(t)$ and complex $g$ setting, shown for both low and high SNR. Higher values indicate better stability for *PIP Corr*, whereas lower values indicate better stability for *PIP MeanDiff* and *Interval SymDiff*; for $\Delta$F1, values closer to zero indicate smaller seed-induced variation.

enlarged cover $\Omega_{J_n}^*(\kappa)^{+c_{\mathrm{loc}}\Delta_{J_n}}$, where $c_{\mathrm{loc}} \geq 1$ is an absolute constant depending only on the locality radius of the spline quasi-interpolant/projection used below.

**Lemma B.1.** *Suppose Assumption 5.1–5.3 holds. Define $\kappa_{J_n} := c_\kappa\, C_\beta J_n^{-\alpha_\beta}$ with $c_\kappa \geq 4$ and $C_\beta$ being some constant. Then for all sufficiently large $n$, we have*

(i) *Uniform approximation at resolution $J_n$:*

$$\|\beta^* - \beta_{J_n}^*\|_\infty \ \leq \ C\, J_n^{-\alpha_\beta} \ \lesssim \ \kappa_{J_n}. \tag{11}$$

*Moreover, on the strong-signal set $\Omega^*(2\kappa_{J_n})$:*

$$\|\beta^* - \beta_{J_n}^*\|_{L^\infty(\Omega^*(2\kappa_{J_n}))} \ \leq \ C_\beta\, J_n^{-\alpha_\beta} \ \leq \ \kappa_{J_n}/c_\kappa, \tag{12}$$

*where $\|v(t)\|_{L^\infty(\Omega^*(2\kappa_{J_n}))} = \sup_{t\in\Omega^*(2\kappa_{J_n})} |v(t)|$.*

(ii) *Localization to the strong-signal cover: There exists an absolute constant $c_{\mathrm{loc}} \geq 1$ and $\Delta_{J_n} \asymp J_n^{-1}$ such that, for all sufficiently large $n$,*

$$\Omega_{J_n}^* \ \subseteq \ \Omega_{J_n}^*(\kappa_{J_n})^{+c_{\mathrm{loc}}\Delta_{J_n}}. \tag{13}$$

(iii) *Index and link truncation: letting $\alpha_1 = \min(\alpha_g, 1)$,*

$$\sup_{X\in\mathcal{X}} |u_{J_n}^*(X) - u^*(X)| \leq C_X \|\beta_{J_n}^* - \beta^*\|_\infty \lesssim J_n^{-\alpha_\beta}, \tag{14}$$

*and*

$$\sup_{X\in\mathcal{X}} |\mu_{J_n}^*(X) - \mu^*(X)| = \sup_{X\in\mathcal{X}} |g^*(u_{J_n}^*(X)) - g^*(u^*(X))| \leq L_g\Big(\sup_{X\in\mathcal{X}} |u_{J_n}^*(X) - u^*(X)|\Big)^{\alpha_1} \lesssim J_n^{-\alpha_\beta\alpha_1}. \tag{15}$$

(iv) *Strong-signal sandwich:*

$$\Omega^*(2\kappa_{J_n}) \subseteq \Omega^*_{J_n} \subseteq \Omega^*_{J_n}(\kappa_{J_n})^{+c_{\mathrm{loc}}\Delta_{J_n}}. \tag{16}$$

(v) *Gap decomposition (no extra structure assumed):*

$$\left|\Omega^*_{J_n} \,\Delta\, \Omega^*(2\kappa_{J_n})\right| \;\leq\; \underbrace{\left|\Omega^*(\kappa_{J_n}) \setminus \Omega^*(2\kappa_{J_n})\right|}_{\text{threshold band}} \;+\; \underbrace{\left|\Omega^*_{J_n}(\kappa_{J_n})^{+c_{\mathrm{loc}}\Delta_{J_n}} \setminus \Omega^*(\kappa_{J_n})\right|}_{\text{resolution boundary layer}}. \tag{17}$$

*Proof.* **Step 1 (spline approximation and the choice of $Q_{J_n}$).** Fix a local B-spline basis $\{B_j\}_{j=1}^{J_n}$ of fixed order on $[0,1]$ with supports $I_j$ and $\Delta_{J_n} := \max_{1 \leq j \leq J_n} \operatorname{diam}(I_j) \asymp J_n^{-1}$. Let $Q_{J_n}$ denote a *local* spline quasi-interpolant/projection onto the corresponding spline space (e.g., a standard quasi-interpolant associated with the B-spline partition). By standard $L_\infty$ spline approximation theory, for any $\beta \in \mathcal{H}^{\alpha_\beta}([0,1])$,

$$\|\beta - Q_{J_n}\beta\|_\infty \leq C_\beta J_n^{-\alpha_\beta}.$$

In the sequel, we work with this fixed operator $Q_{J_n}$.

*Step 2 (localization with an enlarged cover).* Recall $\kappa_{J_n} = c_\kappa C_\beta J_n^{-\alpha_\beta}$. Construct a smooth cut-off function $\chi_{J_n} : [0,1] \to [0,1]$ such that

$$\chi_{J_n}(t) = 1 \text{ for } t \in \Omega^*(2\kappa_{J_n}), \qquad \chi_{J_n}(t) = 0 \text{ for } t \in \Omega^*(\kappa_{J_n})^c,$$

and on the transition band $\Omega^*(\kappa_{J_n}) \cap \Omega^*(2\kappa_{J_n})^c$, one has $0 < \chi_{J_n}(t) < 1$ (with $\chi_{J_n}$ chosen smooth across the boundaries of these sets). Define the truncated coefficient

$$\beta^{\mathrm{loc}}(t) := \beta^*(t)\chi_{J_n}(t).$$

Then $\operatorname{supp}(\beta^{\mathrm{loc}}) \subseteq \Omega^*(\kappa_{J_n})$ and, since $1 - \chi_{J_n}$ vanishes on $\Omega^*(2\kappa_{J_n})$ and is supported inside $\Omega^*(\kappa_{J_n})^c \cup \left(\Omega^*(\kappa_{J_n}) \cap \Omega^*(2\kappa_{J_n})^c\right)$,

$$\|\beta^* - \beta^{\mathrm{loc}}\|_\infty = \sup_{t \in [0,1]} |\beta^*(t)\{1 - \chi_{J_n}(t)\}| \leq \sup_{t \in \Omega^*(\kappa_{J_n})^c} |\beta^*(t)| \leq 2\kappa_{J_n}.$$

Moreover, $\beta^{\mathrm{loc}} \in \mathcal{H}^{\alpha_\beta}([0,1])$ with the same smoothness order, since $\beta^* \in \mathcal{H}^{\alpha_\beta}$ and $\chi_{J_n}$ is smooth and bounded.

Now define

$$\beta^*_{J_n} := Q_{J_n}\beta^{\mathrm{loc}}.$$

By the locality of $Q_{J_n}$, there exists an absolute constant $c_{\mathrm{loc}} \geq 1$ such that

$$\operatorname{supp}(Q_{J_n}f) \subseteq \operatorname{supp}(f)^{+c_{\mathrm{loc}}\Delta_{J_n}} \qquad \text{for all bounded } f.$$

Therefore,

$$\operatorname{supp}(\beta^*_{J_n}) \subseteq \operatorname{supp}(\beta^{\mathrm{loc}})^{+c_{\mathrm{loc}}\Delta_{J_n}} \subseteq \Omega^*(\kappa_{J_n})^{+c_{\mathrm{loc}}\Delta_{J_n}} \subseteq \Omega^*_{J_n}(\kappa_{J_n})^{+c_{\mathrm{loc}}\Delta_{J_n}},$$

where the last inclusion uses $\Omega^*(\kappa_{J_n}) \subseteq \Omega^*_{J_n}(\kappa_{J_n})$. Since $\Omega^*_{J_n} = \operatorname{supp}(\beta^*_{J_n})$ by local linear independence of the B-spline basis, we obtain $\Omega^*_{J_n} \subseteq \Omega^*_{J_n}(\kappa_{J_n})^{+c_{\mathrm{loc}}\Delta_{J_n}}$, which proves (13).

For approximation, note that on $\Omega^*(2\kappa_{J_n})$, we have $\beta^{*\mathrm{loc}} = \beta^*$, hence

$$\|\beta^* - \beta^*_{J_n}\|_{L^\infty(\Omega^*(2\kappa_{J_n}))} = \|\beta^{*\mathrm{loc}} - Q_{J_n}\beta^{*\mathrm{loc}}\|_{L^\infty(\Omega^*(2\kappa_{J_n}))} \leq \|\beta^{*\mathrm{loc}} - Q_{J_n}\beta^{*\mathrm{loc}}\|_\infty \leq C_\beta J_n^{-\alpha_\beta},$$

which gives (12). Globally,

$$\|\beta^* - \beta^*_{J_n}\|_\infty \leq \|\beta^* - \beta^{*\mathrm{loc}}\|_\infty + \|\beta^{*\mathrm{loc}} - Q_{J_n}\beta^{*\mathrm{loc}}\|_\infty \leq 2\kappa_{J_n} + C_\beta J_n^{-\alpha_\beta} \lesssim J_n^{-\alpha_\beta}.$$

This proves (11).

*Step 3 (index and link truncation).* By Assumption 5.1,

$$|u^*_{J_n}(X) - u^*(X)| = \left|\int_0^1 X(t)\big(\beta^*_{J_n}(t) - \beta^*(t)\big)\,dt\right| \leq \|X\|_{L^1}\|\beta^*_{J_n} - \beta^*\|_\infty \leq C_X\|\beta^*_{J_n} - \beta^*\|_\infty,$$

and (14) follows from (11). The link bound (15) follows from Hölder continuity of $g^*$.

*Step 4 (sandwich).* Take $t \in \Omega^*(2\kappa_{J_n})$. Then $|\beta^*(t)| > 2\kappa_{J_n}$ and by (12),

$$|\beta^*_{J_n}(t)| \geq |\beta^*(t)| - \|\beta^* - \beta^*_{J_n}\|_{L^\infty(\Omega^*(2\kappa_{J_n}))} > 2\kappa_{J_n} - \kappa_{J_n}/c_\kappa > 0$$

since $c_\kappa \geq 4$. Hence $t \in \text{supp}(\beta^*_{J_n})$. For standard local B-spline bases (local linear independence), $\text{supp}(\beta^*_{J_n}) = \cup_{j \in S^*_{J_n}} I_j = \Omega^*_{J_n}$, so $\Omega^*(2\kappa_{J_n}) \subseteq \Omega^*_{J_n}$. The right inclusion $\Omega^*_{J_n} \subseteq \Omega^*_{J_n}(\kappa_{J_n})^{+c_{\text{loc}}\Delta_{J_n}}$ follows from (13). This proves (16).

*Step 5 (gap decomposition).* By (16),

$$\Omega^*_{J_n} \Delta \Omega^*(2\kappa_{J_n}) \subseteq \Omega^*_{J_n}(\kappa_{J_n})^{+c_{\text{loc}}\Delta_{J_n}} \setminus \Omega^*(2\kappa_{J_n}) = \left(\Omega^*(\kappa_{J_n}) \setminus \Omega^*(2\kappa_{J_n})\right) \cup \left(\Omega^*_{J_n}(\kappa_{J_n})^{+c_{\text{loc}}\Delta_{J_n}} \setminus \Omega^*(\kappa_{J_n})\right),$$

and (17) follows by subadditivity of Lebesgue measure. □

## B.2. Approximation Error Bounds (Theorem 5.4)

**Lemma B.2.** *Suppose Assumption 5.6 holds such that there exists $L_g$ satisfy $|g^*(u) - g^*(v)| \leq L_g|u - v|^{\alpha_1}$ for all $u, v \in [0, 1]$ with $\alpha_1 = \min(\alpha_g, 1)$. Then for any integer depth $H_n \geq 2$, there exists a univariate ReLU network $f_{H_n}$ with constant width (at most a universal constant) and depth at most $H_n$ such that*

$$\sup_{u \in [0,1]} |f_{H_n}(u) - g^*(u)| \lesssim H_n^{-2\alpha_1}.$$

*Proof.* This result is a direct specialization of Theorem 2 in Yarotsky (2018) to the one–dimensional setting. Taking input dimension $d = 1$ and approximation accuracy $\varepsilon \asymp H_n^{-2\alpha_1}$ in that theorem yields the stated rate with constant network width. □

### B.2.1. PROOF OF THEOREM 5.4

The proof is structured into three steps. First, for the unknown link function $g^*(u)$, Lemma B.2 guarantees the existence of a one-dimensional neural network $f_{H_n-1}$ with constant width and depth $H_n - 1$ such that

$$\sup_{u \in [0,1]} |f_{H_n-1}(u) - g^*(u)| \lesssim H_n^{-2\alpha_1}.$$

Furthermore, the network output $f_{H_n-1}(u^*_{J_n}(X))$, where $u^*_{J_n}(X) = (\boldsymbol{\omega}^*_{J_n})^\top \boldsymbol{\eta}(X)$, can be reinterpreted as an $H_n$-depth network $F_{\boldsymbol{\theta}}$ with $J_n$-dimensional input $\boldsymbol{\eta}(X)$. To construct $F_{\boldsymbol{\theta}}$, we explicitly design its first hidden layer to satisfy the support condition $\text{supp}_{\text{col}}(\boldsymbol{\theta}) = S^*_{J_n}$: we set the width of the first layer as $L_1 \asymp 1$, the weights of the first neuron in the first hidden layer to be $\boldsymbol{W}_{1,1*} = (\boldsymbol{\omega}^*_{J_n})^\top$ and all other row weights $\boldsymbol{W}_{1,j*} = \boldsymbol{0}^\top$ for $j > 1$. The remaining layers then implement $f_{H_n-1}$ acting on the computed scalar $(\boldsymbol{\omega}^*_{J_n})^\top \boldsymbol{\eta}(X)$. This yields a DNN $F_{\boldsymbol{\theta}}$ with depth at most $H_n$, constant width up to universal constants, and $\text{supp}_{\text{col}}(\boldsymbol{\theta}) = S^*_{J_n}$, establishing (i).

Finally, to prove (ii), we decompose the approximation error into a network approximation error and a functional truncation error. For any $X \in \mathcal{X}$,

$$\begin{aligned}
|\mu_{\boldsymbol{\theta}}(X) - \mu^*(X)| &= |F_{\boldsymbol{\theta}}(\boldsymbol{\eta}(X)) - g^*(u^*(X))| \\
&= |f_{H_n-1}(u^*_{J_n}(X)) - g^*(u^*(X))| \\
&\leq \underbrace{|f_{H_n-1}(u^*_{J_n}(X)) - g^*(u^*_{J_n}(X))|}_{I} + \underbrace{|g^*(u^*_{J_n}(X)) - g^*(u^*(X))|}_{II}.
\end{aligned}$$

Term I is bounded directly by Lemma B.2, while term II is controlled via inequality (15) from Lemma B.1. Taking the supremum over $\mathcal{X}$ then yields the final convergence rate stated in (ii).

## B.3. Posterior Consistency (Theorem 5.7)

We first give a general posterior consistency result introduced in Jiang (2007). Specifically, let $D_n = \{(\boldsymbol{x}_i, Y_i)\}_{i=1}^n$ denote the dataset, where $(\boldsymbol{x}_i, Y_i)$ are i.i.d. under the reference distribution $p^*$. Let $\mathcal{P}$ denote the space of probability densities under

consideration. We consider a sequence of model classes (sieves) $P_n \subset \mathcal{P}$, and write $P_n^c = \mathcal{P} \setminus P_n$ for their complements. We construct $P_n$ through a parameter sieve $\Theta_n$ via

$$P_n := \{p_\theta : \theta \in \Theta_n\}.$$

Let $\Pi$ denote the prior measure on $\mathcal{P}$ (or on $\Theta$ via $p_\theta$), and let $\Pi(\cdot \mid D_n)$ denote the corresponding posterior given the data $D_n$. For each $\varepsilon > 0$, define the posterior probability

$$\widehat{\Pi}(\varepsilon) := \Pi(d(p, p^*) > \varepsilon \mid D_n),$$

where the metric $d(\cdot, \cdot)$ denotes the Hellinger distance, defined by $d(p, q) = \sqrt{\int (\sqrt{p} - \sqrt{q})^2}$. Let $N(\varepsilon, P_n, d)$ denote the $\varepsilon$-covering number of $P_n$ with respect to the metric $d$.

**Lemma B.3.** *For a sequence $\varepsilon_n \to 0$, if there exist constants $2 > b > 2b' > 0$ and $t > 0$ such that the following conditions hold for all sufficiently large $n$:*

*(a) $\log N(\varepsilon_n, P_n, d) \le n\varepsilon_n^2$;*

*(b) $\pi(P_n^c) \le \exp(-bn\varepsilon_n^2)$;*

*(c) $\pi\{p \in \mathcal{P} : d_t(p, p^*) \le b'\varepsilon_n^2\} \ge \exp(-b'n\varepsilon_n^2)$,*

*where $d_t(p, p^*) = \frac{1}{t}\left(\int p^*(x)\left(\frac{p^*(x)}{p(x)}\right)^t dx - 1\right)$, then for any $2b' < x < b$, the posterior probability $\widehat{\Pi}(4\varepsilon_n)$ satisfies:*

*(i) $P\left[\widehat{\Pi}(4\varepsilon_n) \ge 2\exp\left(-\frac{1}{2}n\varepsilon_n^2 m(x)\right)\right] \le 2\exp\left(-\frac{1}{2}n\varepsilon_n^2 m(x)\right),$*

*(ii) $\mathbb{E}\left[\widehat{\Pi}(4\varepsilon_n)\right] \le 4\exp\left(-n\varepsilon_n^2 m(x)\right),$*

*where $m(x) := \min\{1, 2 - x, b - x, t(x - 2b')\}$.*

*Proof.* The proof follows from an argument analogous to that of Proposition 1 in Jiang (2007). $\square$

**Lemma B.4.** *Fix any subset $S \subset \{1, \dots, J_n\}$ with $m := |S|$. Consider the proposed DNN $F_\theta$ defined in (4) with input of dimension $J_n$. Let $\theta$ be a network parameter vector with $\|\theta\|_\infty \le E_n$. Let $\tilde\theta$ be another parameter vector such that*

$$\left|W_{1,hg} - \tilde{W}_{1,hg}\right| \le \begin{cases} \delta_1, & g \in S, \\ \delta_2, & g \notin S, \end{cases} \quad 1 \le h \le L_1, \ 1 \le g \le J_n,$$

*and for all deeper-layer coordinates $\mathcal{G}$, including all biases and weights beyond the first layer,*

$$\max_{j \in \mathcal{G}}\left|\theta_j - \tilde\theta_j\right| \le \delta_1.$$

*Then, for all $x \in [-1, 1]^{J_n}$,*

$$\left|F_\theta(x) - F_{\tilde\theta}(x)\right| \le (E_n + \delta_1)^{H_n - 1}\left[H_n(m+1)L_1 \prod_{k=2}^{H_n}(L_{k-1}+1)L_k\, \delta_1\right.$$

$$\left. + \left\{(J_n - m)L_1 \prod_{k=2}^{H_n}(L_{k-1}+1)L_k\right\}\delta_2\right].$$

*Proof.* Consider the pre-activation at the first layer $z_{1,h} = \sum_{g=1}^{J_n} W_{1,hg} x_g + b_{1,h}$. For any $x \in [-1, 1]^{J_n}$, the difference satisfies:

$$|z_{1,h} - \tilde{z}_{1,h}| \le \sum_{g \in S} \delta_1|x_g| + \sum_{g \notin S} \delta_2|x_g| + \delta_1 \le (m+1)\delta_1 + (J_n - m)\delta_2.$$

For $k \geq 2$, let $z_k$ denote the pre-activation vector of layer $k$. Since ReLU is 1-Lipschitz, the error propagates as:

$$\|z_k - \tilde{z}_k\|_\infty \leq \|W_k\|_\infty \|z_{k-1} - \tilde{z}_{k-1}\|_\infty + \|W_k - \tilde{W}_k\|_\infty \|\tilde{z}_{k-1}\|_\infty + \|b_k - \tilde{b}_k\|_\infty.$$

Note that $\|\tilde{a}_{k-1}\|_\infty \leq (E_n + \delta_1)^{k-1}$. By induction over $k = 2, \ldots, H_n$ and accounting for the total number of parameters in each layer (width products $\prod L_k$), the perturbations $\delta_1$ across $H_n$ layers accumulate linearly. The initial perturbation from inactive columns $(J_n - m)\delta_2$ is magnified by the depth-induced factor $(E_n + \delta_1)^{H_n-1}$. Summing these contributions yields the desired bound. $\qquad\square$

### B.3.1. DEFINITION OF $\tilde{M}_{n,1}(\varepsilon_n)$ AND $\tilde{M}_{n,2}(\varepsilon_n)$ IN THEOREM 5.7

- $\sigma_{0,n}^2 \leq \tilde{M}_{n,1}(\varepsilon_n)$;

- $\max\{\sigma^2, \sigma_{0,n}^2, \sigma_{1,n}^2\} \leq \tilde{M}_{n,2}(\varepsilon_n)$.

Here,

$$\tilde{M}_{n,1}(\varepsilon_n) = \min\left\{\frac{(\delta_n')^2}{2\tau A_n + 2\log(4J_n L_1^2)}, \frac{(\omega_n')^2}{2\log(4J_n L_1^2)}\right\}, \tag{18}$$

and

$$\tilde{M}_{n,2}(\varepsilon_n) = \frac{M_n^2}{2\left[b_1 \, n\varepsilon_n^2 + \log(2K_n)\right]}, \tag{19}$$

where $A_n = H_n \log n + H_n \log \bar{L} + \log\{(J_n + 1)L_1\}$,

$$\delta_n' = \frac{c_1 \varepsilon_n}{H_n \, J_n \, L_1 \, (\bar{L})^{2(H_n-1)} (c_0 M_n)^{H_n-1}}, \tag{20}$$

and

$$\omega_n' = \frac{c_1 \varepsilon_n}{J_n \, L_1 \, (\bar{L})^{2(H_n-1)} (c_0 E_n)^{H_n-1}}, \tag{21}$$

with $c_0$ and $c_1$ being some positive constants. In addition, $K_n = (J_n + 1)L_1 + H_n \bar{L}^2$, $\log M_n = O(\log n)$, and that for sufficiently large $n$, $M_n \geq E_n$.

### B.3.2. PROOF OF THEOREM 5.7

We consider the specific scenario developed in Section 3. The observed data is $D_n = \{(x_i(t), Y_i)\}_{i=1}^n$.

Throughout the posterior contraction analysis, we take the reference truth to be the original law

$$p^* := p_{\mu^*},$$

where the model family is indexed by $\mu_\theta(X) := F_\theta(\eta(X))$ and the true mean is $\mu^*(X) = g^*(\langle X, \beta^*\rangle)$. Here, $p_{\mu^*}(\cdot \mid X)$ is the true conditional density of $Y$ given $X$. Let $P_n := \{p_{\mu_\theta} : \theta \in \Theta_n\} \subset \mathcal{P}$ be a sequence of model classes, where $p_{\mu_\theta}(\cdot \mid X)$ is the approximate density induced by the finite-dimensional representation (3) and the sparse DNN defined in (4) and $\Theta_n := \{\theta : \|\theta\|_\infty \leq M_n, C(\theta) \leq k_0 s_n\}$ for a constant $k_0 \geq 2$. Here, $C(\theta) := |S(\theta)|$ with $S(\theta) := \{j \leq J_n : \max_{1 \leq h \leq L_1} |W_{1,hj}| \geq \delta_n'\}$.

To prove Theorem 5.7, according to Lemma B.3, it suffices to verify that for a sequence $\varepsilon_n^2 \lesssim \frac{s_n \log(J_n/s_n)}{n} + \frac{s_n\left(H_n \log n + \log J_n\right)}{n} + \left(H_n^{-2\alpha_1} + J_n^{-\alpha_\beta \alpha_1}\right)^2$, the following conditions hold for all sufficiently large $n$:

(a) $\log N(\varepsilon_n, P_n, d) \leq n\varepsilon_n^2$;

(b) $\pi(P_n^c) \leq \exp(-bn\varepsilon_n^2)$;

(c) $\pi\{p_{\mu_\theta} \in \mathcal{P} : d_t(p_{\mu_\theta}, p_{\mu^*}) \leq b'\varepsilon_n^2\} \geq \exp(-b'n\varepsilon_n^2)$.

**Verification of condition (a).** Set

$$\delta'_n = \frac{c_1 \varepsilon_n}{H_n \, J_n \, L_1 \, (\bar{L})^{2(H_n - 1)} (c_0 M_n)^{H_n - 1}}.$$

Fix $\boldsymbol{\theta} \in \Theta_n$ and let $S := S(\boldsymbol{\theta})$ with $|S| \leq k_0 s_n$. Define the truncated parameter $\boldsymbol{\theta}^{(S)}$ by zeroing out non-activated columns:

$$W_{1,hg}^{(S)} := W_{1,hg} \mathbf{1}\{g \in S\}, \qquad 1 \leq h \leq L_1, \ 1 \leq g \leq J_n,$$

and keep all remaining coordinates unchanged: $\theta_j^{(S)} := \theta_j$ for $j \in \mathcal{G}$ ($\mathcal{G}$ includes all biases and weights beyond the first layer). Then for $g \notin S$,

$$\max_{h \leq L_1} \big|W_{1,hg} - W_{1,hg}^{(S)}\big| = \max_{h \leq L_1} |W_{1,hg}| < \delta'_n.$$

Applying Lemma B.4 with $(\delta_1, \delta_2) = (0, \delta'_n)$ yields

$$\sup_{\boldsymbol{x} \in [-1,1]^{J_n}} |F_{\boldsymbol{\theta}}(\boldsymbol{x}) - F_{\boldsymbol{\theta}^{(S)}}(\boldsymbol{x})| \leq C \, \varepsilon_n, \tag{22}$$

with $C$ being some constant.

Now consider the truncated class

$$\Theta_n^{\text{trunc}}(S) := \Big\{ \boldsymbol{\theta} : \ \|\boldsymbol{\theta}\|_\infty \leq M_n, \boldsymbol{W}_{1,*g} \equiv 0 \text{ for } g \notin S \Big\}.$$

Set

$$\delta_{1,n} := \frac{c_2 \varepsilon_n}{H_n \, (k_0 s_n + 1) \, L_1 \, (\bar{L})^{2(H_n - 1)} (c_0 M_n)^{H_n - 1}}, \qquad c_2 \in (0, 1). \tag{23}$$

Let $\tilde{\boldsymbol{\theta}} \in \Theta_n^{\text{trunc}}(S)$ satisfy the coordinate-wise bounds

$$\max_{g \in S, \, h \leq L_1} |W_{1,hg}^{(S)} - \tilde{W}_{1,hg}| \leq \delta_{1,n}, \qquad \max_{j \in \mathcal{G}} |\theta_j^{(S)} - \tilde{\theta}_j| \leq \delta_{1,n}.$$

For $g \notin S$, we have $W_{1,hg}^{(S)} = \tilde{W}_{1,hg} = 0$, hence the same bound holds with $\delta_2 = 0 \leq \delta'_n$. Therefore Lemma B.4 (with $(\delta_1, \delta_2) = (\delta_{1,n}, \delta'_n)$) and (23) give

$$\sup_{\boldsymbol{x} \in [-1,1]^{J_n}} |F_{\boldsymbol{\theta}^{(S)}}(\boldsymbol{x}) - F_{\tilde{\boldsymbol{\theta}}}(\boldsymbol{x})| \leq C \, \varepsilon_n. \tag{24}$$

Combining (22)–(24),

$$\sup_{\boldsymbol{x} \in [-1,1]^{J_n}} |F_{\boldsymbol{\theta}}(\boldsymbol{x}) - F_{\tilde{\boldsymbol{\theta}}}(\boldsymbol{x})| \leq C \, \varepsilon_n.$$

As $\mu_{\boldsymbol{\theta}}(X) = F_{\boldsymbol{\theta}}(\boldsymbol{x})$ with $\boldsymbol{x} = \boldsymbol{\eta}(X)$, we further have

$$\sup_{X \in \mathcal{X}} |\mu_{\boldsymbol{\theta}}(X) - \mu_{\tilde{\boldsymbol{\theta}}}(X)| \leq C \, \varepsilon_n.$$

Next, since $d^2(p, q) \leq d_0(p, q)$, it suffices to control the Kullback–Leibler divergence. For the Gaussian regression case, the KL divergence is bounded by the squared difference of mean functions. Hence,

$$d_0(p_{\mu_{\boldsymbol{\theta}}}, p_{\mu_{\tilde{\boldsymbol{\theta}}}}) \leq C \, \mathbb{E}_X \big[\mu_{\boldsymbol{\theta}}(X) - \mu_{\tilde{\boldsymbol{\theta}}}(X)\big]^2 \leq C' \varepsilon_n^2,$$

and therefore $d(p_{\boldsymbol{\theta}}, p_{\tilde{\boldsymbol{\theta}}}) \leq C'' \varepsilon_n$ for all large $n$. Consequently, any coordinate-wise $\ell_\infty$–net for $\Theta_n^{\text{trunc}}(S)$ induces a $c \varepsilon_n$–net (for some constant $c > 0$) under $d$. Since $N(\varepsilon; P_n, d)$ is non-increasing in $\varepsilon$, it suffices to bound $N(c \varepsilon_n; P_n, d)$, which is of the same order.

Fix $S \subset \{1, \ldots, J_n\}$ with $m := |S| \leq k_0 s_n$ and let $K_{\text{deep}} := |\mathcal{G}| \leq H_n \bar{L}^2$. Discretise (i) the $m L_1$ active first-layer weights and (ii) the $K_{\text{deep}}$ deep-layer coordinates on a uniform grid over $[-M_n, M_n]$ with mesh width $\delta_{1,n}$. Let $R_n := \left\lceil \frac{2M_n}{\delta_{1,n}} \right\rceil$. By product covering, there exists an $c \varepsilon_n$–net $\mathcal{N}(S)$ of $\{p_{\boldsymbol{\theta}} : \theta \in \Theta_n^{\text{trunc}}(S)\}$ under $d$ such that $|\mathcal{N}(S)| \leq R_n^{m L_1 + K_{\text{deep}}}$.

Since $m \leq k_0 s_n$,

$$\sum_{m=0}^{k_0 s_n} \binom{J_n}{m} \leq \left(\frac{e J_n}{k_0 s_n}\right)^{k_0 s_n}.$$

Let $\mathcal{N} := \bigcup_{|S| \leq k_0 s_n} \mathcal{N}(S)$. For any $\theta \in \Theta_n$, letting $S = S(\theta)$, the truncation bridge and the net $\mathcal{N}(S)$ yield some $\tilde{\theta} \in \mathcal{N}(S) \subset \mathcal{N}$ such that $d(p_\theta, p_{\tilde{\theta}}) \leq \varepsilon_n$. Therefore, for the constant $c > 0$ above,

$$\log N(c\varepsilon_n; P_n, d) \leq k_0 s_n \log \frac{e J_n}{k_0 s_n} + \left(k_0 s_n L_1 + K_{\text{deep}}\right) \log R_n$$

$$\leq k_0 s_n \log \frac{e J_n}{k_0 s_n} + \left(k_0 s_n L_1 + H_n \bar{L}^2\right) \log \frac{2 M_n}{\delta_{1,n}}.$$

Since $N(\varepsilon; P_n, d)$ is non-increasing in $\varepsilon$, $\log N(\varepsilon_n; P_n, d) \leq \log N(c\varepsilon_n; P_n, d)$. By $\log M_n = O(\log n)$ and $\log(1/\varepsilon_n) = O(\log n)$, and noting that $k_0$ is fixed and $k_0 s_n \leq J_n$, we obtain

$$\log N(\varepsilon_n; P_n, d) \lesssim s_n \log \frac{e J_n}{s_n} + \left[s_n L_1 + H_n \bar{L}^2\right]\left(H_n \log n + H_n \log \bar{L} + \log(J_n L_1)\right).$$

By the rate condition of $\varepsilon_n$, the right-hand side is $O(n \varepsilon_n^2)$, hence $\log N(\varepsilon_n; P_n, d) \leq n \varepsilon_n^2$ for all large $n$, verifying condition (a).

**Verification of condition (b).** By the definition of $\Theta_n$, we have

$$\Pi(\Theta_n^c) \leq T_{1,n} + T_{2,n}, \qquad T_{1,n} := \Pi(\|\boldsymbol{\theta}\|_\infty > M_n), \quad T_{2,n} := \Pi(C(\boldsymbol{\theta}) > k_0 s_n).$$

Note that, under the induced prior on densities,

$$\Pi(P_n^c) = \Pi\{\boldsymbol{\theta} \notin \Theta_n\} = \Pi(\Theta_n^c).$$

Denote $\sigma_{\max}^2 = \max\{\sigma^2, \sigma_{0,n}^2, \sigma_{1,n}^2\}$ and $K_n = (J_n + 1) L_1 + H_n \bar{L}^2$. For every coordinate $\theta_j$, $\Pr(|\theta_j| > M_n) \leq 2 \exp\left(-M_n^2 / 2\sigma_{\max}^2\right)$. With the condition (19) on $\sigma_{\max}^2$ and by the union bound, we have

$$T_{1,n} \leq 2 K_n \exp\left(-\frac{M_n^2}{2\sigma_{\max}^2}\right) \leq \exp\left(-b_1 \, n \varepsilon_n^2\right).$$

For each column $g$, define

$$I_g := \mathbf{1}\left\{\max_{1 \leq h \leq L_1} |W_{1,hg}| \geq \delta_n'\right\}, \qquad C(\boldsymbol{\theta}) = \sum_{g=1}^{J_n} I_g.$$

Under the specific prior, the indicators $\{I_g\}_{g=1}^{J_n}$ are i.i.d. Bernoulli $(p_n)$ with

$$p_n := \Pr(I_g = 1) = (1 - \lambda_n) p_{0,n} + \lambda_n p_{1,n} \leq p_{0,n} + \lambda_n, \tag{25}$$

where

$$p_{0,n} := \Pr\left(\max_{1 \leq h \leq L_1} |Z_h| \geq \delta_n'\right), \qquad Z_h \overset{\text{i.i.d.}}{\sim} N(0, \sigma_{0,n}^2),$$

$$p_{1,n} := \Pr\left(\max_{1 \leq h \leq L_1} |W_h| \geq \delta_n'\right), \qquad W_h \overset{\text{i.i.d.}}{\sim} N(0, \sigma_{1,n}^2).$$

By the union bound and Gaussian tails,

$$p_{0,n} \leq 2 L_1 \exp\left(-\frac{(\delta_n')^2}{2\sigma_{0,n}^2}\right), \qquad p_{1,n} \leq 1.$$

Under assumption (18) for $\sigma_{0,n}^2$, we have

$$\frac{(\delta_n')^2}{2\sigma_{0,n}^2} \geq \tau A_n + \log(4 J_n L_1^2),$$

with $A_n = H_n \log n + H_n \log \bar{L} + \log\{(J_n + 1)L_1\}$, and hence

$$p_{0,n} \leq 2L_1 \exp\left(-\frac{(\delta'_n)^2}{2\sigma_{0,n}^2}\right) \leq 2L_1 \exp\left\{-\tau A_n - \log(4J_n L_1^2)\right\} = \frac{1}{2J_n L_1} e^{-\tau A_n}. \tag{26}$$

Therefore $J_n p_{0,n} \leq \frac{1}{2L_1} e^{-\tau A_n}$, which is $o(s_n)$ and in particular implies $J_n p_{0,n} \leq \frac{1}{4} k_0 s_n$ for all large $n$. Moreover, under Assumption 5.5, we have $J_n \lambda_n \lesssim \left[(n\bar{L})^{H_n}(J_n + 1)L_1\right]^{-\tau'}$, so for all large $n$,

$$J_n \lambda_n \leq \frac{1}{4} k_0 s_n.$$

Combining the last two displays yields

$$J_n p_n \;\leq\; J_n(p_{0,n} + \lambda_n) \;\leq\; \frac{1}{2} k_0 s_n.$$

Let $q_n := k_0 s_n / J_n$ and assume $q_n \leq 1/2$ for all large $n$. Since $\{I_g\}_{g=1}^{J_n}$ are i.i.d. Bernoulli$(p_n)$, we have

$$C(\boldsymbol{\theta}) = \sum_{g=1}^{J_n} I_g \;\sim\; \text{Bin}(J_n, p_n).$$

Under the above bounds, we have $p_n \leq q_n/2$, hence $q_n > p_n$. Applying Zubkov & Serov (2013, Theorem 1) to $X \sim \text{Bin}(J_n, p_n)$ with $k = \lfloor J_n q_n \rfloor - 1$ yields

$$\Pr\{C(\boldsymbol{\theta}) \geq k_0 s_n\} \;\leq\; 1 - \Phi\left(\sqrt{2J_n\, H(p_n, q_n)}\right),$$

where

$$H(p_n, q_n) = q_n \log\frac{q_n}{p_n} + (1 - q_n)\log\frac{1 - q_n}{1 - p_n} = \text{KL}(q_n \| p_n).$$

Using the standard Gaussian tail bound $1 - \Phi(t) \leq e^{-t^2/2}$ for $t > 0$, we obtain

$$\Pr\{C(\boldsymbol{\theta}) \geq k_0 s_n\} \;\leq\; \exp\left\{-J_n\,\text{KL}(q_n \| p_n)\right\}.$$

Moreover, since $p_n \leq q_n/2$ and $\log(q_n/p_n) \to \infty$ in our regime, the negative term $(1 - q_n)\log\{(1 - q_n)/(1 - p_n)\}$ is negligible compared to $q_n \log(q_n/p_n)$, and in particular for all sufficiently large $n$,

$$\text{KL}(q_n \| p_n) \;\geq\; \frac{1}{2}\, q_n \log\frac{q_n}{p_n}.$$

Therefore,

$$\Pr\{C(\boldsymbol{\theta}) \geq k_0 s_n\} \;\leq\; \exp\left\{-c\, k_0 s_n \log\frac{q_n}{p_n}\right\}$$

for some absolute constant $c > 0$.

Moreover, under Assumption 5.5 and (25) and (26), we can ensure $p_n \lesssim e^{-\tau A_n}/J_n$ for some $\tau > 0$, so that

$$\log\frac{q_n}{p_n} \;\gtrsim\; \tau A_n + \log(k_0 s_n) \;\gtrsim\; A_n \quad \text{for all large } n.$$

Therefore,

$$-\log \Pr\{C(\boldsymbol{\theta}) \geq k_0 s_n\} \;\gtrsim\; k_0 s_n A_n.$$

Then, under Assumption 5.6, we have $n\varepsilon_n^2 \lesssim s_n A_n$, and thus for some $b_2 > 0$,

$$T_{2,n} = \Pi\{C(\boldsymbol{\theta}) > k_0 s_n\} \leq \exp(-b_2\, n\varepsilon_n^2),$$

verifying condition (b).

**Verification of condition (c).** We check condition (c) for $t = 1$. Consider the set

$$\mathcal{A}_n = \Big\{ \boldsymbol{\theta} : \max_{g \in S^*_{J_n}} \max_{h \leq L_1} |W_{1,hg} - W^*_{1,hg}| \leq \omega_n, \ \max_{g \notin S^*_{J_n}} \max_{h \leq L_1} |W_{1,hg}| \leq \omega'_n, \ \|\boldsymbol{\theta}_{\mathcal{G}} - \boldsymbol{\theta}^*_{\mathcal{G}}\|_\infty \leq \omega_n \Big\},$$

where $\omega_n = \frac{c_1 \varepsilon_n}{[H_n(s_n+1)L_1(\bar{L})^{2(H_n-1)}(c_0 E_n)^{H_n-1}]}$ and $\omega'_n$ are defined in (20) and (21), and $\boldsymbol{\theta}^*$ is the network parameter vector of the DNN $F_{\boldsymbol{\theta}^*}$ obtained in Theorem 5.4. If $\boldsymbol{\theta} \in \mathcal{A}_n$, then by Lemma B.4 we have

$$\sup_{X \in \mathcal{X}} |\mu_{\boldsymbol{\theta}}(X) - \mu_{\boldsymbol{\theta}^*}(X)| = \sup_{\boldsymbol{x} \in [-1,1]^{J_n}} |F_{\boldsymbol{\theta}}(\boldsymbol{x}) - F_{\boldsymbol{\theta}^*}(\boldsymbol{x})| \leq 3c_1 \varepsilon_n,$$

where $\boldsymbol{x} = \boldsymbol{\eta}(X)$ and $\mu_{\boldsymbol{\theta}}(X) = F_{\boldsymbol{\theta}}(\boldsymbol{\eta}(X))$. Define

$$\tilde{\xi}_n := \inf_{\boldsymbol{\theta}: \, \text{supp}_{\text{col}}(\boldsymbol{\theta}) = S^*_{J_n}, \, \|\boldsymbol{\theta}\|_\infty \leq E_n} \sup_{X \in \mathcal{X}} |\mu_{\boldsymbol{\theta}}(X) - \mu^*(X)|.$$

Then, Theorem 5.4 gives $\tilde{\xi}_n \lesssim H_n^{-2\alpha_1} + J_n^{-\alpha_\beta \alpha_1}$.

Since

$$\sup_{X \in \mathcal{X}} |\mu_{\boldsymbol{\theta}^*}(X) - \mu^*(X)| \leq \tilde{\xi}_n,$$

we have,

$$\sup_{X \in \mathcal{X}} |\mu_{\boldsymbol{\theta}}(X) - \mu^*(X)| \leq 3c_1 \varepsilon_n + \tilde{\xi}_n.$$

For normal models, we obtain

$$d_1(p_{\mu_{\boldsymbol{\theta}}}, p_{\mu^*}) \leq C(1 + o(1)) \mathbb{E}_X [\mu_{\boldsymbol{\theta}}(X) - \mu^*(X)]^2 \leq C(1 + o(1))(3c_1 \varepsilon_n + \tilde{\xi}_n)^2, \quad \text{if } \boldsymbol{\theta} \in \mathcal{A}_n,$$

for some constant $C$. Under Assumption 5.6, we have $n\varepsilon_n^2 \geq M_0 \, n \, \tilde{\xi}_n^2$ for large $M_0$. Thus for any small $b' > 0$, condition (c) holds as long as $c_1$ is sufficiently small, and the prior satisfies $-\log \Pi(\mathcal{A}_n) \leq b' \, n\varepsilon_n^2$.

Let $S^*_{J_n} \subset \{1, \ldots, J_n\}$ be the true active column set with $|S^*_{J_n}| = s_n$, and define the configuration $\boldsymbol{\gamma}^*$ by $\gamma^*_g = \mathbf{1}\{g \in S^*_{J_n}\}$. Write $K_{\text{deep}} := |\mathcal{G}| \leq H_n \bar{L}^2$.

Since columns are independent under the hierarchical prior and $(\gamma_g)$ are i.i.d.,

$$\Pi(\mathcal{A}_n) \geq \Pi(\boldsymbol{\gamma} = \boldsymbol{\gamma}^*) \, \Pi(\mathcal{A}_n \mid \boldsymbol{\gamma} = \boldsymbol{\gamma}^*).$$

To bound $\Pi(\mathcal{A}_n)$ from below, we consider the event where the selection indicators match the target indices exactly, i.e., $\gamma_g = 1$ if $g \in S^*_{J_n}$ and $\gamma_g = 0$ otherwise.

*(i) Consider* $\Pi(\boldsymbol{\gamma} = \boldsymbol{\gamma}^*)$.

$$\Pi(\boldsymbol{\gamma} = \boldsymbol{\gamma}^*) = \lambda_n^{s_n}(1 - \lambda_n)^{J_n - s_n}, \quad \Rightarrow \quad -\log \Pi(\boldsymbol{\gamma} = \boldsymbol{\gamma}^*) \leq s_n \log \frac{1}{\lambda_n} + (J_n - s_n)\lambda_n.$$

Under Assumption 5.5, $(J_n - s_n)\lambda_n = o(n\varepsilon_n^2)$ and $s_n \log(1/\lambda_n) \lesssim s_n\{H_n \log n + H_n \log \bar{L} + \log(J_n L_1)\}$.

*(ii) Consider* $\Pi(\mathcal{A}_n \mid \boldsymbol{\gamma} = \boldsymbol{\gamma}^*)$.

For $X \sim N(0, \sigma^2)$ and any $|a| \leq E_n$,

$$\Pr(|X - a| \leq \omega) \geq 2\omega \cdot \inf_{|u-a| \leq \omega} \phi(u; 0, \sigma^2) \geq c \frac{\omega}{\sigma} \exp\Big(-\frac{(E_n + \omega)^2}{2\sigma^2}\Big),$$

hence

$$-\log \Pr(|X - a| \leq \omega) \lesssim \log \frac{\sigma}{\omega} + \frac{(E_n + 1)^2}{2\sigma^2}.$$

Applying this bound to the active first-layer weights ($s_n L_1$ coordinates with slab variance $\sigma_{1,n}^2$) and to the deep parameters ($K_{\text{deep}}$ coordinates with variance $\sigma^2$), we obtain

$$-\log \Pi(\mathcal{A}_n \mid \gamma = \gamma^*) \leq C \left[ s_n L_1 \left\{ \log \frac{\sigma_{1,n}}{\omega_n} + \frac{(E_n + 1)^2}{2\sigma_{1,n}^2} \right\} + K_{\text{deep}} \left\{ \log \frac{\sigma}{\omega_n} + \frac{(E_n + 1)^2}{2\sigma^2} \right\} \right]$$
$$- \log \Pi \left( \max_{g \notin S_{J_n}^*} \max_{h \leq L_1} |W_{1,hg}| \leq \omega_n' \mid \gamma_g = 0 \; \forall g \notin S_{J_n}^* \right).$$

For the inactive part, by (19) and (21), we have

$$\frac{(\omega_n')^2}{2\sigma_{0,n}^2} \geq \log(4 J_n L_1^2),$$

so for $Z \sim N(0, \sigma_{0,n}^2)$,

$$\Pr(|Z| > \omega_n') \leq 2 \exp\left( -\frac{(\omega_n')^2}{2\sigma_{0,n}^2} \right) \leq \frac{1}{2 J_n L_1^2}.$$

Hence, for each $(g, h)$ with $g \notin S_{J_n}^*$,

$$\Pr(|W_{1,hg}| \leq \omega_n' \mid \gamma_g = 0) \geq 1 - \frac{1}{2 J_n L_1^2}.$$

By independence over $(g, h)$,

$$\Pi \left( \max_{g \notin S_{J_n}^*} \max_{h \leq L_1} |W_{1,hg}| \leq \omega_n' \mid \gamma_g = 0 \; \forall g \notin S_{J_n}^* \right) \geq \left( 1 - \frac{1}{2 J_n L_1^2} \right)^{(J_n - s_n) L_1} \geq e^{-1}$$

for all large $n$.

Combining the above bounds and using $K_n \leq H_n \bar{L}^2$ and $\log(1/\omega_n) = O\big( H_n \log n + H_n \log \bar{L} + \log(s_n L_1) \big)$, we conclude that

$$-\log \Pi(\mathcal{A}_n) \leq C' \left\{ s_n \log \frac{1}{\lambda_n} + \left[ s_n L_1 + H_n \bar{L}^2 \right] \big( H_n \log n + H_n \log \bar{L} + \log(s_n L_1) \big) \right\} \leq b' n \varepsilon_n^2,$$

where the last inequality follows from the rate condition and the hyper-parameter bounds in Assumption 5.5. Consequently, $\Pi(\mathcal{A}_n) \geq \exp(-b' n \varepsilon_n^2)$, verifying condition (c).

## B.4. Selection Consistency (Theorem 5.9)

Denote $q_j = \Pi(r_j = 1 | D_n) = \mathbb{E}(r_j | D_n)$ and $A_n(\varepsilon_n) = \big\{ \boldsymbol{\theta} : \; d\big( p_{\mu_{\boldsymbol{\theta}}}, p_{\mu^*} \big) \geq \varepsilon_n \big\}$.

### B.4.1. PROOF OF THEOREM 5.9

Fix $j \in \{1, \ldots, J_n\}$. By the definition of $q_j$, we have

$$|q_j - e_j^*| = \left| \mathbb{E}((\gamma_j - e_j^*)|D_n) \right| \leq \mathbb{E}\left( |\gamma_j - e_j^*| \big| D_n \right).$$

Split according to $A_n(4\varepsilon_n)$:

$$\mathbb{E}\left( |\gamma_j - e_j^*| \big| D_n \right) \leq \mathbb{E}\left[ |\gamma_j - e_j^*| \mathbf{1}\{\theta \notin A_n(4\varepsilon_n)\} \big| D_n \right] + \mathbb{E}\left[ |\gamma_j - e_j^*| \mathbf{1}\{\theta \in A_n(4\varepsilon_n) \big| D_n \} \right].$$

Since $|\gamma_j - e_j^*| \leq 1$, the second term is bounded by $\Pi\big( A_n(4\varepsilon_n) \mid D_n \big)$, while the first term is controlled by $\rho_n(4\varepsilon_n)$ in Assumption 5.8. Taking the maximum over $j$ gives

$$\max_j |q_j - e_j^*| \leq \rho_n(4\varepsilon_n) + \Pi\big( A_n(4\varepsilon_n) \mid D_n \big) \xrightarrow{P} 0.$$

Then based on Theorem 2.3 stated in Sun et al. (2022), with an appropriate choice of prior hyperparameters, the estimated $\hat{q}_j = Pr(r_j = 1|\hat{\theta})$ based on the MAP estimate $\hat{\theta}$ and $q_j$ are approximately the same as $n \to \infty$. Thus, $\hat{q}_j$ is also a consistent estimator of $e_j^*$, which proves (i). Parts (ii) and (iii) follow immediately from (i).

To prove (iv), recall that the estimated active region on the original domain is $\widehat{\Omega} := \bigcup_{j \in \widehat{S}_{1/2}} I_j$, and the population (truncated) active region at resolution $J_n$ is $\Omega_{J_n}^* = \bigcup_{j \in S_{J_n}^*} I_j$. By Parts (i)–(iii), $\Pr(\widehat{S}_{1/2} = S_{J_n}^*) \to 1$, hence also $\Pr(\widehat{\Omega} = \Omega_{J_n}^*) \to 1$. Therefore, it suffices to show $|\Omega_{J_n}^* \Delta \Omega^*| \to 0$.

By the triangle inequality for symmetric differences,

$$|\Omega_{J_n}^* \Delta \Omega^*| \leq |\Omega_{J_n}^* \Delta \Omega^*(2\kappa_{J_n})| + |\Omega^*(2\kappa_{J_n})\Delta\Omega^*|. \tag{27}$$

For the first term, Lemma B.1(v) gives

$$|\Omega_{J_n}^* \Delta \Omega^*(2\kappa_{J_n})| \leq |\Omega^*(\kappa_{J_n}) \setminus \Omega^*(2\kappa_{J_n})| + \left|\Omega_{J_n}^*(\kappa_{J_n})^{+c_{\mathrm{loc}}\Delta_{J_n}} \setminus \Omega^*(\kappa_{J_n})\right|.$$

The first summand is controlled by Assumption 5.2, since $\Omega^*(\kappa_{J_n}) \setminus \Omega^*(2\kappa_{J_n}) \subseteq \{t \in \Omega^* : |\beta^*(t)| \leq 2\kappa_{J_n}\}$ and $2\kappa_{J_n} \downarrow 0$, hence $|\Omega^*(\kappa_{J_n}) \setminus \Omega^*(2\kappa_{J_n})| \leq |\{t \in \Omega^* : |\beta^*(t)| \leq 2\kappa_{J_n}\}| \to 0$.

For the second summand, we relate $\Omega_{J_n}^*(\kappa)$ to $\Omega^*(\kappa)$ via a $\Delta_{J_n}$-enlargement. Indeed, by definition $\Omega^*(\kappa) \subseteq \Omega_{J_n}^*(\kappa)$; moreover, since each spline support interval $I_j$ has diameter at most $\Delta_{J_n}$, if $t \in I_j$ for some $j$ with $I_j \cap \Omega^*(\kappa) \neq \varnothing$, then there exists $s \in I_j \cap \Omega^*(\kappa)$ such that $|t - s| \leq \mathrm{diam}(I_j) \leq \Delta_{J_n}$, implying $\Omega_{J_n}^*(\kappa) \subseteq \Omega^*(\kappa)^{+\Delta_{J_n}}$. Consequently,

$$\Omega_{J_n}^*(\kappa)^{+c_{\mathrm{loc}}\Delta_{J_n}} \subseteq \left(\Omega^*(\kappa)^{+\Delta_{J_n}}\right)^{+c_{\mathrm{loc}}\Delta_{J_n}} = \Omega^*(\kappa)^{+(c_{\mathrm{loc}}+1)\Delta_{J_n}},$$

and hence

$$\left|\Omega_{J_n}^*(\kappa_{J_n})^{+c_{\mathrm{loc}}\Delta_{J_n}} \setminus \Omega^*(\kappa_{J_n})\right| \leq \left|\Omega^*(\kappa_{J_n})^{+(c_{\mathrm{loc}}+1)\Delta_{J_n}} \setminus \Omega^*(\kappa_{J_n})\right|.$$

Since $\Omega^*(\kappa_{J_n})$ is a finite union of intervals, its boundary has finite cardinality; therefore there exists $C_\partial > 0$ such that for all $\delta > 0$, $\left|\Omega^*(\kappa_{J_n})^{+\delta} \setminus \Omega^*(\kappa_{J_n})\right| \leq C_\partial \delta$. Taking $\delta = (c_{\mathrm{loc}} + 1)\Delta_{J_n}$ and using $\Delta_{J_n} \asymp J_n^{-1} \to 0$ yields $\left|\Omega_{J_n}^*(\kappa_{J_n})^{+c_{\mathrm{loc}}\Delta_{J_n}} \setminus \Omega^*(\kappa_{J_n})\right| \to 0$. This shows $|\Omega_{J_n}^* \Delta \Omega^*(2\kappa_{J_n})| \to 0$.

For the second term in (27), note that $\Omega^* \setminus \Omega^*(2\kappa_{J_n}) \subseteq \{t \in \Omega^* : |\beta^*(t)| \leq 2\kappa_{J_n}\}$, and hence

$$|\Omega^*(2\kappa_{J_n})\Delta\Omega^*| = |\Omega^* \setminus \Omega^*(2\kappa_{J_n})| \leq \left|\{t \in \Omega^* : |\beta^*(t)| \leq 2\kappa_{J_n}\}\right| \to 0$$

by Assumption 5.2. Combining the above bounds gives $|\Omega_{J_n}^* \Delta \Omega^*| \to 0$, and therefore

$$|\widehat{\Omega}\Delta\Omega^*| \leq |\widehat{\Omega}\Delta\Omega_{J_n}^*| + |\Omega_{J_n}^* \Delta \Omega^*| \xrightarrow{P} 0,$$

which completes the proof of (iv).

## C. Additional Simulation Analysis

### C.1. Simulation Details and Supplementary Results (Data from Section 6.1)

Following the functional-covariate generation mechanism in AdaFNN, we generate $X_i(\cdot)$ from a truncated cosine expansion on $\mathcal{T} = [0, 1]$. Let $\phi_1(t) \equiv 1$ and $\phi_k(t) = \sqrt{2}\cos\left((k-1)\pi t\right)$ for $k = 2, \ldots, K$ with $K = 50$, and set $X_i(t) = \sum_{k=1}^{K} c_{ik}\phi_k(t)$. We draw $c_{ik} = z_k r_{ik}$ with $r_{ik} \overset{\text{i.i.d.}}{\sim} \mathrm{Unif}(-\sqrt{3}, \sqrt{3})$ and $z_1 = 20$, $z_2 = z_3 = 15$, and $z_k = 1$ for $k \geq 4$. We observe each curve on a discrete grid over $[0, 1]$.

We consider three localized settings for $\beta(\cdot)$, which we label as *Simple*, *Medium*, and *Complex* according to increasing difficulty in region recovery. Let $T(t; a, b) := (t - a)(b - t)\mathbf{1}\{t \in [a, b]\}$ and let $\tilde{T}(t; a, b)$ denote its normalized version on $[a, b]$ (so that $\max_{t \in [a,b]} \tilde{T}(t; a, b) = 1$). Thus, on each active interval, $\beta(\cdot)$ has a smooth quadratic bump that vanishes at the endpoints, and multiple active intervals are represented by sums over disjoint bumps. Specifically,

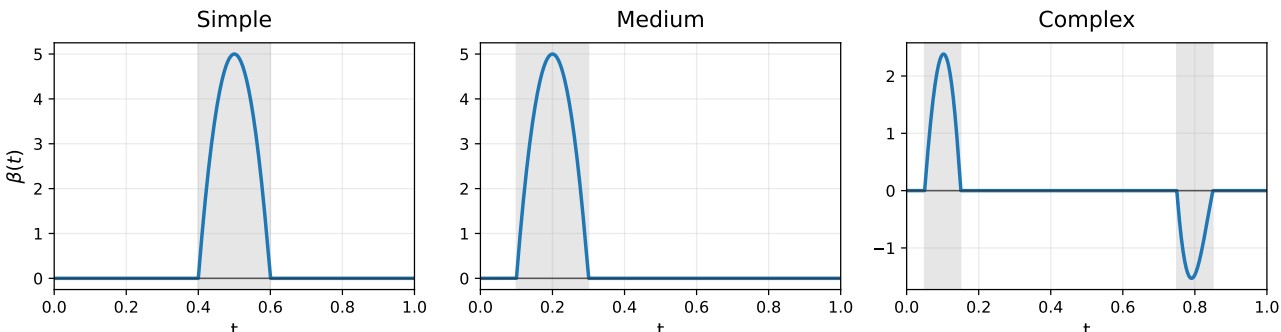

*Figure 7.* Illustration of the three types of coefficient functions $\beta(t)$ used in simulations: Simple (single interior bump), Medium (single boundary bump), and Complex (two separated oscillating bumps).

we use: (i) (**Simple**) a single centered bump on $[0.4, 0.6]$, $\beta(t) = 5\tilde{T}(t; 0.4, 0.6)$; (ii) (**Medium**) a single boundary-adjacent bump on $[0.1, 0.3]$, $\beta(t) = 5\tilde{T}(t; 0.1, 0.3)$; and (iii) (**Complex**) two separated narrow bumps with within-region oscillation. Let $W = [a_1, b_1] \cup [a_2, b_2]$ with $(a_1, b_1) = (0.05, 0.15)$ and $(a_2, b_2) = (0.75, 0.85)$, and set $\beta(t) = 2.5 \sum_{m=1}^{2} \tilde{T}(t; a_m, b_m) \sin(2\pi(t + 0.1))$. The shapes of the three $\beta(t)$ types are illustrated in Figure 7.

To vary the nonlinearity of the response mechanism, we consider four choices of the link function $g^*$ with increasing complexity: (i) linear $g^*(u) = u$; (ii) logistic-type $g^*(u) = \{1 + \exp(u)\}^{-1}$; (iii) sinusoidal $g^*(u) = \sin(u)$; and (iv) a composite link $g^*(u) = \tanh(u) + \sin(4u) \exp(-0.01u^2)$.

Finally, given a draw of $X_i(\cdot)$ and a choice of $(\beta, g^*)$, we generate the response as

$$Y_i = g^*\left(\int_0^1 X_i(t)\beta(t)\,dt\right) + \varepsilon_i, \qquad \varepsilon_i \overset{\text{i.i.d.}}{\sim} \mathcal{N}(0, \sigma_\varepsilon^2).$$

To make noise levels comparable across settings, we calibrate additive Gaussian noise by a target signal-to-noise ratio, $\text{SNR} = \text{Var}(\text{signal})/\text{Var}(\text{noise})$. In all simulations, the latent curves $X_i(\cdot)$ generate the responses above, but the learning algorithms observe only discretely sampled noisy curves obtained by adding i.i.d. Gaussian measurement noise at $\text{SNR} = 10$ on the observation grid. The response noise variance $\sigma_\varepsilon^2$ is chosen so that the resulting responses attain the target response SNR (we consider $\text{SNR} \in \{5, 10\}$) in each scenario. When required, we apply a denoising step to the noisy curve observations before constructing spline features. For each scenario, we simulate 100 replicates and summarize the results in Figures 2, 4, 8, and 9.

## C.2. Additional Simulation Analysis with Lower SNR

To further evaluate the proposed method under more challenging noise conditions, we conduct an additional analysis focusing on low-SNR scenarios (SNR = 2 and 3). Both prediction metrics and region-selection metrics, reported as mean (SD) over the 50 replications, are shown in Table 9 for the scenario with composite link function $g$ and various settings of $\beta(t)$. As expected, all methods become more challenging, yet our approach remains stable and continues to outperform the baselines.

## C.3. Additional Simulation Analysis with Model-Misspecification and Non-Gaussian Noise

To further assess robustness beyond the main simulation settings, we consider two additional scenarios. The first is an FGAM-type mean misspecification setting: $Y_i^{\text{clean}} = \int_0^1 F(X_i(t), t)\,dt$ where $F(x, t) = -0.5 + \exp\left[-\left(\frac{x}{5.0}\right)^2 - \left(\frac{t - 0.5}{0.3}\right)^2\right]$, and the second is a heavy-tailed error setting with Student-$t$ noise (degrees of freedom = 3) under the original single-index mean with simple $\beta(t)$ and composite link function $g$. For both scenarios, we evaluate performance at SNR = 5 and SNR = 10. Tables 10 and 11 report the corresponding prediction and selection results as mean(standard deviation) over these 50 replications.

Under the FGAM mean misspecification setting, sBayFDNN achieves the best prediction performance at both SNR levels and also delivers the strongest support recovery overall. In particular, it attains the highest F1 among the competing

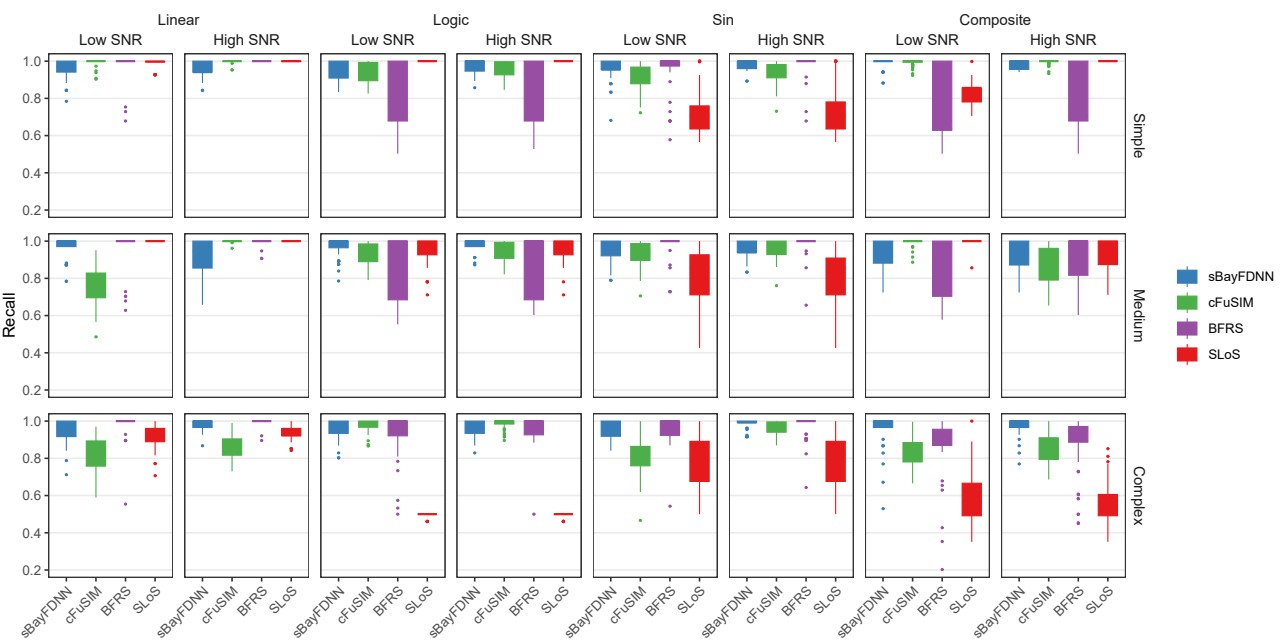

*Figure 8.* Recall values across $g$ functions, SNR settings, and $\beta(t)$ scenarios.

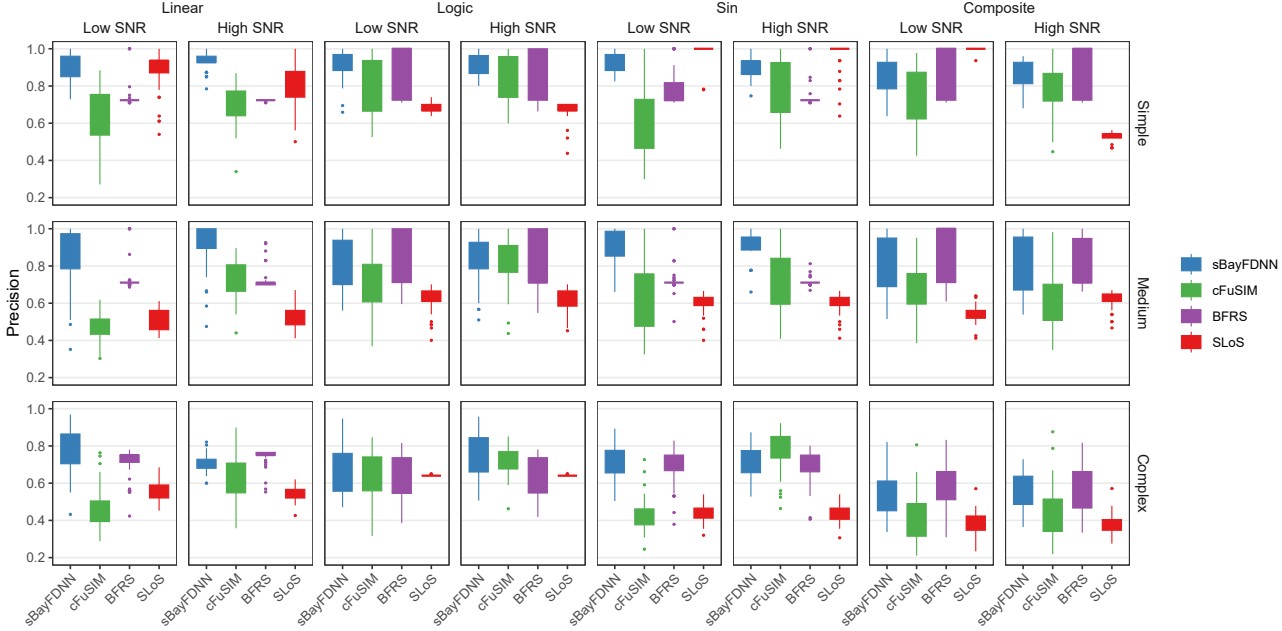

*Figure 9.* Precision values across $g$ functions, SNR settings, and $\beta(t)$ scenarios.

*Table 9.* Simulation results under scenarios with composite $g$ and lower SNR levels. Prediction and selection metrics are reported as mean(standard deviation).

| SNR | $\beta(t)$ Setting | Method | RMSE | F1 | Recall | Precision |
|---|---|---|---|---|---|---|
| | | sBayFDNN | **1.019(0.071)** | **0.641(0.066)** | 0.811(0.238) | **0.601(0.169)** |
| | | FNN | 1.100(0.108) | – | – | – |
| | Complex | AdaFNN | 1.175(0.209) | – | – | – |
| | | cFuSIM | 1.023(0.068) | 0.488(0.053) | 0.766(0.124) | 0.358(0.079) |
| | | BFRS | 1.073(0.081) | 0.517(0.268) | 0.503(0.307) | 0.583(0.270) |
| | | SLoS | 1.072(0.078) | 0.621(0.060) | **0.912(0.106)** | 0.472(0.044) |
| | | sBayFDNN | **0.853(0.042)** | **0.850(0.081)** | 0.964(0.060) | 0.771(0.132) |
| | | FNN | 0.949(0.070) | – | – | – |
| 2.0 | Medium | AdaFNN | 0.887(0.144) | – | – | – |
| | | cFuSIM | 1.113(0.324) | 0.606(0.049) | **1.000(0.000)** | 0.437(0.048) |
| | | BFRS | 0.946(0.051) | 0.735(0.260) | 0.721(0.298) | **0.798(0.310)** |
| | | SLoS | 0.944(0.055) | 0.578(0.094) | 0.735(0.174) | 0.493(0.111) |
| | | sBayFDNN | 0.866(0.058) | **0.924(0.053)** | 0.979(0.028) | 0.882(0.095) |
| | | FNN | 0.944(0.054) | – | – | – |
| | Simple | AdaFNN | **0.855(0.171)** | – | – | – |
| | | cFuSIM | 0.883(0.116) | 0.855(0.077) | **0.988(0.027)** | 0.754(0.124) |
| | | BFRS | 0.959(0.065) | 0.767(0.098) | 0.691(0.194) | 0.930(0.117) |
| | | SLoS | 0.960(0.065) | 0.901(0.032) | 0.831(0.049) | **0.986(0.044)** |
| | | sBayFDNN | **0.908(0.063)** | **0.651(0.068)** | 0.818(0.127) | 0.549(0.069) |
| | | FNN | 0.998(0.080) | – | – | – |
| | Complex | AdaFNN | 0.964(0.185) | – | – | – |
| | | cFuSIM | 0.926(0.078) | 0.530(0.071) | 0.805(0.080) | 0.395(0.090) |
| | | BFRS | 0.974(0.076) | 0.562(0.256) | 0.606(0.323) | **0.556(0.225)** |
| | | SLoS | 0.972(0.068) | 0.609(0.057) | **0.912(0.102)** | 0.459(0.047) |
| | | sBayFDNN | 0.742(0.040) | **0.843(0.079)** | 0.964(0.060) | **0.761(0.131)** |
| | | FNN | 0.845(0.064) | – | – | – |
| 3.0 | Medium | AdaFNN | **0.729(0.070)** | – | – | – |
| | | cFuSIM | 0.839(0.245) | 0.636(0.137) | 0.835(0.129) | 0.540(0.189) |
| | | BFRS | 0.847(0.052) | 0.724(0.257) | 0.741(0.305) | 0.748(0.292) |
| | | SLoS | 0.847(0.051) | 0.455(0.082) | **1.000(0.000)** | 0.298(0.070) |
| | | sBayFDNN | 0.753(0.055) | **0.908(0.055)** | **0.989(0.023)** | 0.844(0.093) |
| | | FNN | 0.837(0.058) | – | – | – |
| | Simple | AdaFNN | 0.802(0.187) | – | – | – |
| | | cFuSIM | **0.734(0.062)** | 0.662(0.119) | **0.989(0.029)** | 0.497(0.137) |
| | | BFRS | 0.864(0.064) | 0.826(0.032) | 0.787(0.141) | 0.910(0.124) |
| | | SLoS | 0.865(0.062) | 0.874(0.088) | 0.839(0.075) | **0.947(0.168)** |

*Table 10.* Simulation results for the stress setting with FGAM mean under two noise levels. Prediction and selection metrics are reported as mean(standard deviation).

| SNR | Method | RMSE | F1 | Recall | Precision |
|---|---|---|---|---|---|
| | sBayFDNN | **0.049(0.006)** | **0.907(0.196)** | **0.873(0.268)** | **1.000(0.000)** |
| | FNN | 0.049(0.004) | – | – | – |
| 5 | AdaFNN | 0.051(0.002) | – | – | – |
| | cFuSIM | 0.051(0.003) | 0.824(0.006) | 0.701(0.009) | **1.000(0.000)** |
| | BFRS | 0.053(0.003) | 0.000(0.000) | 0.000(0.000) | 0.000(0.000) |
| | SLoS | 0.053(0.003) | 0.000(0.000) | 0.000(0.000) | 0.000(0.000) |
| | sBayFDNN | **0.040(0.006)** | **0.806(0.246)** | **0.736(0.331)** | **1.000(0.000)** |
| | FNN | 0.041(0.003) | – | – | – |
| 10 | AdaFNN | 0.049(0.002) | – | – | – |
| | cFuSIM | 0.042(0.006) | 0.700(0.061) | 0.542(0.074) | **1.000(0.000)** |
| | BFRS | 0.051(0.003) | 0.000(0.000) | 0.000(0.000) | 0.000(0.000) |
| | SLoS | 0.051(0.003) | 0.000(0.000) | 0.000(0.000) | 0.000(0.000) |

*Table 11.* Simulation results for the stress setting with non-Gaussian error, simple $\beta(t)$, and composite link function $g$ under two noise levels. Prediction and selection metrics are reported as mean(standard deviation).

| SNR | Method | RMSE | F1 | Recall | Precision |
|---|---|---|---|---|---|
| 5 | sBayFDNN | **0.658(0.077)** | **0.921(0.044)** | 0.953(0.053) | **0.900(0.092)** |
| | FNN | 0.704(0.084) | – | – | – |
| | AdaFNN | 0.796(0.295) | – | – | – |
| | cFuSIM | 0.661(0.093) | 0.736(0.138) | 0.988(0.035) | 0.602(0.169) |
| | BFRS | 0.777(0.076) | 0.802(0.054) | 0.852(0.196) | 0.811(0.137) |
| | SLoS | 0.777(0.077) | 0.625(0.021) | **1.000(0.000)** | 0.455(0.022) |
| 10 | sBayFDNN | **0.553(0.066)** | **0.944(0.025)** | 0.973(0.038) | **0.921(0.061)** |
| | FNN | 0.614(0.067) | – | – | – |
| | AdaFNN | 0.579(0.055) | – | – | – |
| | cFuSIM | 0.585(0.063) | 0.751(0.125) | 0.989(0.032) | 0.620(0.158) |
| | BFRS | 0.712(0.066) | 0.830(0.045) | 0.854(0.166) | 0.858(0.150) |
| | SLoS | 0.713(0.065) | 0.389(0.039) | **1.000(0.001)** | 0.242(0.030) |

region-selection methods, indicating that the proposed approach remains effective even when the true mean structure departs from the assumed single-index form. The results therefore suggest a useful degree of robustness to mean misspecification.

Under the heavy-tailed error setting, sBayFDNN again performs strongly across both prediction and selection metrics. It achieves the best or nearly best prediction accuracy and yields the highest F1 and precision among the region-selection competitors. Although performance differences are naturally smaller in some prediction metrics, the overall pattern indicates that the proposed method remains robust when the Gaussian noise assumption is violated by heavy-tailed disturbances.

## D. Details for the Real-world Datasets

**Implementation details.** Relative to the simulation defaults in Appendix A.1, we keep the same network architecture and use mini-batch size 32. For smaller-sample real datasets, we recommend a less aggressive first-layer sparsification prior to mitigate underfitting; accordingly, we use $(\lambda_n, \sigma_{0,n}^2, \sigma_{1,n}^2) = (10^{-1}, 10^{-5}, 5 \times 10^{-2})$ for Tecator, Bike, and IHPC. (Unless otherwise stated, other hyperparameters follow Appendix A.1.)

**ECG.** The ECG dataset is from the EchoNext dataset on PhysioNet (Elias & Finer, 2025; Goldberger et al., 2000) and downloaded from https://physionet.org/content/echonext/1.1.0/. To reduce phase variability, we detect R-peaks and align each waveform to the detected R-peak, extract a fixed-length beat-centered window, and resample it to a common grid of length $L = 256$ at 250 Hz, following standard ECG preprocessing practice (Kachuee et al., 2018; Makowski et al., 2021). To better approximate an i.i.d. sample, we remove repeated measurements from the same subject; after deduplication, the final sample sizes are 26,192/4,618/5,434 for train/validation/test, equal to the numbers of unique patients in each split. The window length is selected using the training split only by screening candidate pre/post windows and choosing the configuration that minimizes variability of the aligned R-peak location across subjects (trimmed standard deviation below 0.03 on the normalized phase), which yields a symmetric window of 0.3 s before and 0.3 s after the R-peak.

To assess region identification, we define silver-standard intervals on the original domains and then map them to the normalized domain $\mathcal{T} = [0, 1]$ induced by our fixed-window. For ECG, we use a 120 ms window (Yu et al., 2003; Hummel et al., 2009) centered at the R peak, i.e., $[-0.06, 0.06]$ seconds relative to the R peak, as a silver-standard proxy for the QRS complex extent.

To adjust for scalar covariates, we residualize the response by fitting an OLS regression on the training split (age, sex, acquisition year, location setting, and race/ethnicity) and applying the fitted adjustment to validation/test splits. For ECG, we use a higher-capacity network (7 hidden layers of width 512; total depth 8) trained with mini-batch size 512 and learning rate $5 \times 10^{-4}$; we consider $\mathcal{J} = \{180, 200, 220\}$ and use B-splines of degree 8 to accommodate the larger sample size and sharply localized QRS morphology. Here, a larger $J_n$ is used because the signal contains a narrow peak; smaller $J_n$ would yield overly broad selections. Table 12 reports the ECG sensitivity results over different spline resolutions $J_n$; all metrics are computed as described in Appendix A.4. It is observed that the selected set of candidate $J_n$ values achieves a good balance between recall and precision.

**Tecator.** The Tecator dataset is downloaded from https://lib.stat.cmu.edu/datasets/tecator. It contains near-infrared ab-

*Table 12.* Results for ECG dataset under different $J_n$ settings.

| $J_n$ | MeanLen | Curve Roughness | RMSE | F1 | Recall | Precision | MinLen/Mesh |
|-----|---------|-----------------|---------|--------|--------|-----------|-------------|
| 20  | 1.0000  | 0.0000          | 14.2331 | 0.3333 | 1.0000 | 0.2000    | 20.0000     |
| 40  | 1.0000  | 0.0410          | 12.8696 | 0.3333 | 1.0000 | 0.2000    | 40.0000     |
| 60  | 1.0000  | 0.1559          | 12.3688 | 0.3333 | 1.0000 | 0.2000    | 60.0000     |
| 80  | 1.0000  | 0.1063          | 12.3784 | 0.3333 | 1.0000 | 0.2000    | 80.0000     |
| 100 | 0.2935  | 0.1818          | 12.1960 | 0.5083 | 1.0000 | 0.3407    | 1.0870      |
| 120 | 0.1845  | 0.2235          | 12.1353 | 0.5308 | 1.0000 | 0.3613    | 1.0714      |
| 140 | 0.1591  | 0.1899          | 12.0042 | 0.5906 | 1.0000 | 0.4190    | 1.0606      |
| 160 | 0.1066  | 0.1660          | 12.1815 | 0.5458 | 1.0000 | 0.3753    | 1.0526      |
| 180 | 0.1570  | 0.1251          | 11.9560 | 0.6359 | 1.0000 | 0.4758    | 20.9302     |
| 200 | 0.0955  | 0.1296          | 11.9265 | 0.6039 | 0.7344 | 0.4216    | 9.3750      |
| 220 | 0.1486  | 0.1233          | 12.0542 | 0.5979 | 0.8302 | 0.4156    | 22.8302     |
| 240 | 0.0733  | 0.0887          | 12.0183 | 0.4764 | 0.5000 | 0.4549    | 9.3103      |

sorbance spectra of 240 meat samples measured on 100 wavelength channels ranging from 850 nm to 1,050 nm, together with moisture (water), fat and protein percentages determined by analytic chemistry. We use water as the response and follow the official split (129 training, 43 monitoring/validation, and 43 testing samples). Relative to the simulation defaults in Appendix A.1, we keep the same network architecture but use mini-batch size 32. We use $\mathcal{J} = \{80, 100, 120\}$. We use the water-related absorption band around 970–980 nm(van Kollenburg et al., 2021) and define the silver-standard wavelength interval as $[965, 985]$ nm. The interval is then mapped to the normalized domain $\mathcal{T} = [0, 1]$ via wavelength normalization.

**Bike.**   We use the Bike Sharing dataset (Fanaee-T, 2013) in its hourly-resolution form and represent each day as a functional observation with $T = 24$ equally spaced time points on $\mathcal{T} = [0, 1]$. The dataset can be obtained from https://archive.ics.uci.edu/dataset/275/bike%2Bsharing%2Bdataset. We define the response as the total demand over the next 7 days and adopt a chronological train/validation/test split with sample sizes $453/97/98$ (total $N = 648$), covering the date range 2011-01-16 to 2012-12-30 (train end: 2012-06-12; validation end: 2012-09-17). We use $\mathcal{J} = \{8, 10, 12, 14\}$.

**IHPC.**   We use the Individual Household Electric Power Consumption (IHPC) dataset from the UCI Machine Learning Repository https://archive.ics.uci.edu/dataset/235/individual+household+electric+power+consumption (Hebrail & Berard, 2006). We use the minute-averaged global active power trajectory as the functional input, yielding daily curves of length $T = 1440$ on $\mathcal{T} = [0, 1]$. After restricting to complete days and constructing next-day prediction pairs, we obtain $N = 1290$ samples and use a chronological train/validation/test split with sample sizes $672/329/289$, spanning 2006-12-17 to 2010-11-24.We use $\mathcal{J} = \{15, 20, 30, 40\}$.

Table 13 reports runtime on the IHPC dataset. Under this high-resolution setting, sBayFDNN remains practically feasible, with an average runtime of about 674 seconds on the reported hardware configuration. Although sBayFDNN is not the fastest method overall, but it is faster than BFRS and substantially more practical than the available CPU-only FNN implementation and cFuSIM under default dense-grid settings. This suggests that the proposed projection-based framework remains computationally feasible for high-resolution functional inputs, even when repeated fitting is incorporated.

*Table 13.* IHPC runtime (mean (sd), seconds) comparison across methods under (AMD Ryzen 5 5600H CPU, NVIDIA RTX 3050 Ti Laptop GPU with 4GB VRAM, and 16GB RAM).

|         | sBayFDNN        | AdaFNN          | FNN      | cFuSim   | BFRS            | SLoS             |
|---------|-----------------|-----------------|----------|----------|-----------------|------------------|
| Runtime | 673.641 (27.851)| 513.336 (23.642)| >20 min  | >30 min  | 819.805 (37.151)| 104.144 (12.219) |

