# OpenReview forum: "Sparse Bayesian Deep Functional Learning with Structured Region Selection"
_ICML.cc/2026/Conference — ICML 2026 regular_

### Official Review · Reviewer_r89h · 2026-03-10

**Soundness:** 3
**Presentation:** 2
**Significance:** 2
**Originality:** 1
**Overall Recommendation:** 4
**Confidence:** 2

**Summary:**

This paper proposes an empirical "Bayesian" deep neural network for sparse scalar-on-function regression. The method projects a functional predictor onto a B-spline basis, applies a variable selection "spike-and-slab" prior to these projections, and then uses the selected first-layer column norms to identify "active" regions of the functional domain with the union of the supports of the respective B-spline basis functions. The paper contains approximation, posterior contraction, and selection-consistency results, along with simulation studies and four real-data examples.

**Compliance With Llm Reviewing Policy:**

Affirmed.

**Final Justification:**

See rebuttal comment.

**Key Questions For Authors:**

1. What is the exact model class being fit in Sections 3 and 4: a single-index model $g(<X, \beta>)$ after spline truncation, or a general DNN on spline scores $\eta_j(X)$? Please rewrite this part precisely. If the intended fitted model is the much larger DNN class, how/why/to what extent would the theoretical guarantees developed in Section 5 apply to it?
2. How should the selected "active regions" be interpreted under the known identifiability issues of scalar-on-function regression, especially when the functional predictor is low-rank and/or strongly collinear across nearby locations? Please explain what assumptions are needed for region-level interpretability in practice, and why the Tecator results should be viewed as a "successful" support recovery rather than resulting in one of many admissible surrogates/solutions.
3. Why should the MAP plug-in quantities in Section 4.2 be regarded as meaningful posterior inclusion probabilities? Have the authors checked stability of the selected regions across many random initializations, or compared against any sampler-based approximation on smaller problems to verify that the chosen mode is representative?

**Limitations:**

No. The paper briefly mentions future extensions, but it does not adequately discuss the current method's main limitations: reliance on a single functional predictor with Gaussian loss, the two-stage residualization used in the ECG example, the discrepancy between local MAP fitting and fully Bayesian inference, the fragility/impossibility of "active region" interpretation under functional non-identifiability issues, the very limited scope of the simulation study, the gap between their theoretical development for single index models and their use of a much more general DNN model.

**Strengths And Weaknesses:**

Strengths

- Relevant problem. Interpretable nonlinear scalar-on-function regression with some notion of region selection seems useful for applications such as ECG and spectroscopy.
- Empirical results suggest that the proposed estimator can be predictive in the considered benchmarks, and the paper compares against several relevant functional baselines.
- Manuscript is ambitious in trying to provide a theoretical analysis of the deep function approximation and region selection in their framework.

Weaknesses

- ~~Soundness: serious mismatch between stated model and fitted estimator. Equation (1) presents a functional single-index model of the form $g(<X, \beta>)$, and Equation (3) still has that structure after spline truncation. However, Equation (4) then defines the fitted model as a general DNN acting on the vector of spline scores $\eta(X)$. This is a much larger model class than a single-index model. The theory in Section 5 and Appendix B is completely written around the single-index structure with a sparse coefficient function beta, but the optimization in Section 4 fits a generic DNN on spline features. As written, the theoretical guarantees do not apply to or justify the full fitted procedure nor do they justify the paper's broad claims about "the first theoretical guarantees for a Bayesian deep functional model." The model class the theory is being developed for does not fit the model actually being fitted.~~
- Soundness: selection and interpretability claims are based on strong identifiability assumptions that are not adequately addressed IMO. In regression with functional covariates, coefficient functions and selected regions are generally not identifiable without additional constraints, especially when X(t) is effectively low-rank (in a nutshell: $\int x(t)\beta(t)dt = \int x(t)(\beta(t) + n_x(t))dt$ for any function $n_x(t)$ in the null space of the covariance operator of X(t), see e.g. DOI: 10.1214/16-EJS1123). The paper's Assumption 5.8 is explicitly an identifiability assumption, but neither the practical implications nor how realistic that assumption is are discussed. On real data, the Tecator example seems to show exactly this this problem rather than resolve it: the method gets recall 1.0 but precision is only 0.204, and the paper gives a post hoc explaination for the extra selected regions via collinearity or compositional coupling.
- Soundness: the uncertainty-quantification claims are very strongly overstated. The method does not perform posterior inference over model space in any substantive Bayesian sense. Section 4 computes a local MAP estimate, then defines "posterior inclusion probabilities" that define the active regions by plugging in the MAP first-layer weights. These are not posterior marginal inclusion probabilities integrating over a (almost always highly(!!) multimodal) spike-and-slab posterior.
In Appendix A.2, predictions are averaged across restarts, but metrics are reported only for the single restart with the best validation loss. This makes the region-selection output potentially very initialization-sensitive, without any discussion of the stability of these results in the paper.
- Soundness: the evidence calculation in Appendix A.2 is not convincing as Bayesian model selection. The paper uses a Laplace approximation around a single local mode, after post-sparsifying the fitted network, in a highly non-convex and likely very multimodal problem. This looks more like a heuristic score than principled Bayesian evidence, and the paper does not validate that it selects stable or optimal solutions.
- Soundness: the simulation study is limited and mostly "home-court". All data-generating models come exactly from the paper's own single-index data-generating mechanism with Gaussian errors. There are no nonlinear scalar-on-function truths of the FGAM type, no multiple-functional-predictor settings, no non-Gaussian outcomes, no investigation of sensitivity of results to different hyperparameters for the various priors, i.e.: no stress tests at all. This is not enough to support the paper's broad empirical claims.
- Presentation: the methodology section is confusing and difficult to assess because it blurs together three very different objects: the assumed population single-index model, its spline truncation/representation, and the much more flexible DNN model actually being optimized. The paper needs to separate more clearly between what is assumed for theory's sake, what model (class) is being fit in practice, and what exactly is being selected/interpreted in their MAP-finding implementation.
- Presentation: the related-work discussion is incomplete. In particular, richer nonlinear scalar-on-function models such as functional generalized additive models of the form \int F(X(t), t) dt (see e.g. 10.1080/10618600.2012.729985) are not discussed, even though they are directly relevant when motivating the need for nonlinear functional regression beyond linear FLR. Also missing is a discussion of related work on sparse scalar-on-function regression models such as arxiv.org/pdf/1811.00577 or sparse functional data embeddings like e.g. doi.org/10.1080/01621459.2015.1016225.
- Significance: the application scope is narrower than the framing suggests. The current formulation handles a single functional predictor with $L_2$ (gaussian) loss. In the ECG study, scalar covariates are handled by residualizing the outcome first; this undermines coherent interpretation and uncertainty quantification for the combined problem.  The vonclusion does acknowledge some of these limitations, but they drastically reduce the practical reach of the proposal IMO.
- Originality: while all the elements being used here are all well-established in prior work, there is some originality in the combination of spline features, first-layer structured sparsity, and a deep network for this kind of model. However, because the paper's theoretical and empirical claims are not aligned with the actual fitting procedure or fitted model class, I do not think the current version establishes a  clear or sound novel contribution.

---

> ### Author Rebuttal · Authors · 2026-03-31
>
> Thank you very much for your constructive feedback. We will clarify theory‑model alignment and non‑identifiability, justify MAP estimation, and expand simulations, related work, and limitations. Detailed results (Responses 3–5) are available at  https://osf.io/3gehx/overview?view_only=af6c51d11d304c58aa8210657e58a04f
> due to space constraints.
>
> 1. Soundness 1: We would like to clarify that the theoretical framework is fully aligned with the fitted model. Specifically, the DNN $F_{\theta}$ directly estimates $h$, making the single‑index structure a special case of the DNN model class. Our goal is not to recover the exact single‑index architecture, but to find a DNN $F_{\theta}$ that accurately estimates the conditional expectation $E(Y\mid X(t))$ under the true single‑index model, which is a standard approach in nonparametric statistical learning. The theoretical analysis accounts for this: Thm. 5.4 shows that there exists a DNN $F_{\theta}$ with the correct sparse column support such that the approximation error $|F_{\theta}(\eta(X(t))-g(\int X(t)\beta(t)dt)|$ decays at the specified rate ($\mu_{\theta}$ and $\mu*$ are introduced for the two terms (P5), it does not require recovering the exact single‑index architecture, only that given $X(t)$, $\mu*$ is well‑approximated by $\mu_{\theta}$. This is sufficient, as posterior consistency (Thm. 5.7) directly considers the Hellinger distance between the true conditional distribution $p_{\mu^*}$ from the single‑index model and the DNN‑induced distribution $p_{\mu_{\theta}}$, and selection consistency (Thm. 5.9) follows accordingly.
>
> 2. Soundness 2: We acknowledge the non-identifiability issue, which motivates Assumption 5.8. It rules out predictively near-equivalent but structurally distinct sparse solutions. It is more plausible when the covariance structure of the functional predictor provides sufficient local distinguishability, and the Spike-and-Slab prior further regularizes toward a sparse, localized representative. Under strong local collinearity, however, the assumption is harder to assess, and exact support recovery should be interpreted with caution. This aligns with the Tecator results: even after preprocessing, nearby wavelengths remain correlated, and several additionally selected bands are correlated with the silver interval. Thus, the low precision is partly explained by correlation-equivalent surrogate bands rather than purely arbitrary false positives. Accordingly, the selected regions are better viewed as a data-supported predictive band family.
>
> 3. Soundness 3: We agree that full Bayesian sampling is more common, but MAP‑based PIPs are a well‑established approach in sparse Bayesian modeling (Ročková, 2018 AOS; Ročková and George, 2018 JASA; Gan et al., 2019 JASA; Yang et al., 2021 JMLR), offering computational efficiency and theoretical guarantees. While the true posterior may be multimodal, Thm. 5.9 ensures that MAP‑based PIPs converge to the true selection indicators. To verify finite‑sample stability, we conducted experiments across different initializations and found region selections to be highly consistent. To ensure consistency, we will report region‑selection results based on the average of five restarts, as with predictions, which are nearly identical to those from the best‑validation restart. We will clarify that the uncertainty quantification pertains specifically to region selection rather than all model parameters.
>
> 4-5. Soundness 4-5: We agree that the quantity is not exact Bayesian evidence. It serves as a Laplace‑type surrogate for comparing resolutions and restarts within a candidate set, following prior Bayesian studies (Sun et al., 2022, JASA). We have added an empirical check showing that this criterion selects stable and near‑optimal $J_n$ values. We have expanded the simulation study accordingly, and our method still shows competitive performance.
>
> 6-7. Presentations 1-2: We will restructure Section 3 around Eq. (1)–(4) to clearly illustrate the connections among the three models and clarify that all theoretical results are established for the DNN-based model under the single‑index truth. The three references will be added as suggested.
>
> 8. Significance: We acknowledge the limitations of the two‑step approach, though it remains a common preprocessing step in biomedical applications (Frésard et al., 2019, Nat. Med.). Our main contribution is a sparse Bayesian deep model for a single functional predictor and scalar response—a broadly relevant setting that lays the groundwork for future extensions. As noted in Conclusion, integrating scalar covariates directly is a clear direction for future work, which we will emphasize.
>
> 9. Originality: With the theory‑model alignment clarified above, we believe the novelty of both the methodology and the theoretical contributions is well justified.
>
> 10-12. Q1-Q3: Please see our response to Soundness 1-3.
>
> 13. Limitations: We will expand Conclusion to cover these aspects.

---

> > ### Author Rebuttal · Reviewer_r89h · 2026-04-02
> >
> > I appreciate the detailed and patient reply and the additional benchmarks.
> > I see that my claim that the theoretical framework is not aligned with the fitted model was based on a misunderstanding.
> > I will adjust my scoring accordingly.

---

> > > ### Author Response · Authors · 2026-04-03
> > >
> > > We sincerely thank you for the careful reconsideration and constructive feedback throughout the discussion. We greatly appreciate your updated assessment, and we will take your suggestions into account in revising the manuscript.

---

### Official Review · Reviewer_juT9 · 2026-03-11

**Soundness:** 4
**Presentation:** 4
**Significance:** 4
**Originality:** 4
**Overall Recommendation:** 5
**Confidence:** 3

**Summary:**

This paper proposes a Bayesian functional deep neural network that combines nonlinear predictive modeling for functional inputs with structured region selection driven by a spike-and-slab prior on first-layer weights. The method outputs posterior inclusion probabilities to identify influential functional domains and claims theoretical guarantees plus strong empirical performance across simulations and real datasets. Overall the paper is well written and the proposed method is novel and solid.

**Compliance With Llm Reviewing Policy:**

Affirmed.

**Key Questions For Authors:**

I have a few questions that might help clarify the methodology and improve reproducibility.

First, the inference procedure seems to rely on MAP estimation with plug-in posterior inclusion probabilities. I’m curious how well this approximates full posterior uncertainty in practice, especially for regions that are borderline in terms of selection.

Second, it would be helpful if the paper provided clearer guidance on default hyperparameter choices such as prior variances and the threshold  𝜏 along with some sensitivity analysis showing how these settings affect region recovery.

Finally, since the paper mentions code in the supplementary materials, it would be great to know whether the authors plan to release it publicly upon acceptance and whether it will include instructions to reproduce all of the experiments.

**Limitations:**

yes

**Strengths And Weaknesses:**

Soundness: The model is clearly defined, combining a functional basis representation with a deep neural network and a Bayesian prior for structured region selection. The paper presents both theoretical guarantees and empirical validation to show their model is sound. Results on real datasets show competitive or best performance in terms of RMSE and MAE. I particularly found this is very userful that the selected functional regions are illustrated through posterior inclusion probabilities for examples such as ECG and spectral data.

Presentation: very good presentation.

Significance: The work is highly relevant for functional data analysis and interpretable deep learning, particularly in scientific applications such as ECG analysis, neuroimaging, and wearable sensing.

Originality: Integrating Bayesian deep functional modeling with structured region selection and theoretical guarantees is a meaningful contribution.

---

> ### Author Rebuttal · Authors · 2026-03-31
>
> Thank you very much for your thorough and highly encouraging review. We are grateful for your recognition of the novelty, soundness, and significance of our work, and for your valuable suggestions to further improve the manuscript. In response to your three questions, we will (1) add a discussion on the practical behavior of MAP‑based PIPs, supported by Theorem 5.9 and Figure 3, to clarify their role in uncertainty quantification; (2) provide a sensitivity analysis and clearer guidance on hyperparameter choices; and (3) confirm our plan to publicly release the code with detailed documentation upon acceptance to ensure full reproducibility. We believe these additions will further strengthen the clarity and impact of our work. Please refer to our detailed one-by-one responses below.
>
> 1. Q1: Thank you for your insightful comment. We agree that full Bayesian sampling is more commonly used for uncertainty quantification. However, MAP‑based PIPs are a well‑established approach in sparse Bayesian modeling (Ročková, 2018 AOS; Ročková and George, 2018 JASA; Gan et al., 2019 JASA; Yang et al., 2021 JMLR), offering computational efficiency while maintaining theoretical guarantees. Theorem 5.9 shows that these plug‑in quantities converge to the true selection indicators under mild conditions, ensuring consistency. Figure 3 illustrates the estimated PIPs in a simulated setting alongside the true signal; the results demonstrate that the PIPs effectively capture the probability of selection—regions with stronger signals are more reliably selected, and the behavior near region boundaries remains well‑calibrated. Moreover, the strong F1 scores and other metrics based on the selected regions further support the practical validity of the approach. We will add a discussion of these points in the revised manuscript.
>
> 2. Q2: Thank you for your suggestion. The default hyperparameter settings, including the B‑spline degree, number of knots, network architecture, and prior variances, are already described in Appendix A.1 and D. For the threshold $\tau$, we set it to 0.5 by default. Following your suggestion, we have conducted a sensitivity analysis and found that the region recovery results remain stable across a reasonable range of these hyperparameters. We will add this analysis to the revised manuscript to provide clearer guidance on practical choices.
>
> Sensitivity analysis of hyperparameters (macro-averaged over scenarios; mean (sd)), with one hyperparameter varied at a time while the others are fixed at their default values.
>
> $\lambda_n$      RMSE        F1        Recall        Precision
>
> 1.0e-06  0.6747(0.1180),  0.7868(0.1472),  0.7574(0.1582),  0.8349(0.1261);
>
> 3.0e-06 0.6813(0.1273), 0.8182(0.2748), 0.7859(0.2667), 0.8532(0.1764);
>
> 1.0e-05 0.6693(0.1201), 0.7982(0.1071), 0.8884(0.1045), 0.7346(0.1007);
>
> 3.0e-05 0.6928(0.1303), 0.7095(0.2294), 0.6742(0.3097), 0.8470(0.0529);
>
> 1.0e-04 0.6743(0.1183), 0.7947(0.1382), 0.7746(0.1438), 0.8315(0.1228).
>
> $\sigma_0^2$ RMSE        F1        Recall        Precision
>
> 1.0e-06  0.6542(0.1271),  0.5463(0.0754),  0.9990(0.0010),  0.3826(0.0732);
>
> 3.0e-06  0.6928(0.1303), 0.7095(0.2294), 0.6742(0.3097), 0.8470(0.0529);
>
> 1.0e-05  0.6693(0.1201), 0.7982(0.1071), 0.8884(0.1045), 0.7346(0.1007);
>
> 3.0e-05  0.6937(0.1229), 0.6684(0.2178), 0.5682(0.2576), 0.8952(0.0752);
>
> 1.0e-04  0.6873(0.1232), 0.6991(0.1973), 0.6642(0.2592), 0.7380(0.0322).
>
> $\sigma_1^2$  RMSE        F1        Recall        Precision
>
> 1.0e-03   0.6920(0.1195),  0.7035(0.1536),  0.6346(0.2215),  0.7892(0.0238);
>
> 1.5e-03   0.6832(0.1256), 0.7150(0.2056), 0.6785(0.3121), 0.8606(0.0016);
>
> 2.0e-03   0.6693(0.1201), 0.7982(0.1071), 0.8884(0.1045), 0.7346(0.1007);
>
> 3.0e-03   0.6928(0.1303), 0.7095(0.2294), 0.6742(0.3097), 0.8470(0.0529);
>
> 5.0e-03   0.6534(0.1278), 0.7759(0.1444), 0.9854(0.0028), 0.6696(0.2009).
>
> $\tau$   F1             Recall         Precision     $n_{selected}$
>
> 0.1   0.8256(0.0217) 0.9930(0.0031) 0.7251(0.0316) 11.51(0.48)
>
> 0.2   0.8969(0.0137) 0.9573(0.0083) 0.8579(0.0244)  8.30(0.21)
>
> 0.3   0.8969(0.0137) 0.9573(0.0083) 0.8579(0.0244)  8.30(0.21)
>
> 0.4   0.8781(0.0126) 0.8610(0.0164) 0.9146(0.0201)  5.83(0.19)
>
> 0.5   0.8781(0.0126) 0.8610(0.0164) 0.9146(0.0201)  5.83(0.19)
>
> 0.6   0.7428(0.0281) 0.6498(0.0340) 0.9201(0.0159)  3.32(0.22)
>
> 0.7   0.7428(0.0281) 0.6498(0.0340) 0.9201(0.0159)  3.32(0.22)
>
> 0.8   0.4557(0.0407) 0.3523(0.0353) 0.6864(0.0497)  1.20(0.13)
>
> 0.9   0.4557(0.0407) 0.3523(0.0353) 0.6864(0.0497)  1.20(0.13)
>
> 3. Q3: Thank you for your question. The code has already been uploaded as supplementary material on OpenReview. It includes the complete implementation of our method, the code for generating simulation data, and the scripts for producing the relevant figures and evaluation metrics. Usage instructions are provided within the code. Upon acceptance, we plan to release the code publicly on GitHub, along with more detailed documentation to facilitate reproducibility of all experiments.

---

> > ### Author Rebuttal · Reviewer_juT9 · 2026-04-03
> >
> > Thanks the authors for the additional results. I remain my positive score.

---

> > > ### Author Response · Authors · 2026-04-03
> > >
> > > We sincerely thank you for the positive and constructive feedback throughout the discussion. We greatly appreciate your time and your continued positive assessment.

---

### Official Review · Reviewer_xutz · 2026-03-12

**Soundness:** 3
**Presentation:** 2
**Significance:** 2
**Originality:** 2
**Overall Recommendation:** 4
**Confidence:** 4

**Summary:**

This paper proposes a sparse Bayesian functional deep neural network (sBayFDNN) to address complex nonlinear modeling and interpretable region selection in Functional Data Analysis (FDA). The method projects continuous functions onto B-spline bases to form finite-dimensional features and introduces a structured spike-and-slab prior in the first layer of the DNN to achieve region-level domain selection via Posterior Inclusion Probabilities (PIPs). Theoretically, the authors establish guarantees for approximation error bounds, posterior contraction rates, and region selection consistency. Empirically, the method’s superiority in predictive accuracy and interpretability is validated on datasets such as ECG monitoring and spectrometry.

**Compliance With Llm Reviewing Policy:**

Affirmed.

**Final Justification:**

The authors have addressed my key concerns in the response. I have revised my score to weak accept.

**Key Questions For Authors:**

Q1: How does the choice of basis function resolution J_n specifically impact predictive accuracy and region identification precision? Is there a criterion for automatically selecting the optimal J_n?
Q2: Is the "minimum signal strength (beta-min)" condition assumed in Theorem 5.9 easily satisfied in real-world datasets like Tecator? How robust is the model’s PIP performance when signals are weak?
Q3: In the ECG experiment, the precision reported is relatively low (0.464). Does this imply that the selected regions are generally wider than the actual QRS interval? How do the authors interpret the clinical implications of this "over-inclusion"?
Q4: Given that MAP estimation requires multiple random restarts, what is the training time cost on large-scale, high-resolution data such as IHPC?

**Limitations:**

The model exhibits dependence on the configuration of B-spline bases (e.g., degree and knot placement). Furthermore, the current framework primarily handles univariate functional inputs; its robustness in processing multi-source, heterogeneous functional data (especially in collinear scenarios) requires further exploration. In clinical settings, inaccurate region selection could potentially mislead decision-making, a risk that warrants more explicit discussion in the paper.

**Strengths And Weaknesses:**

Strengths：
1.	Clear and important motivation: Simultaneously addressing nonlinear modeling and sparse region selection in functional data is a genuine and significant problem. The paper is well-motivated by practical applications such as ECG monitoring and neuroimaging.
2.	Innovative method design: Combining deep neural networks with Bayesian functional data analysis, and achieving sparse region selection through structured priors (region-level spike-and-slab type priors), is genuinely novel in the existing literature. The design of representing the continuous functional domain through a library of locally supported B-spline bases and introducing binary selection variables at the region level is well-conceived.
3.	Systematic theoretical contributions: The paper establishes theoretical guarantees on three fronts — approximation error bounds, posterior contraction rates, and region selection consistency. The authors claim this is the first work to provide theoretical guarantees for Bayesian deep functional models, which, if true, is a meaningful contribution.
4.	Comprehensive experiments: The paper includes experiments under various simulation settings (linear, nonlinear, different sparsity patterns) and on multiple real-world datasets, with comparisons to multiple baseline methods (e.g., FNN, AdaFNN, cFuSIM, BFRS, and SLoS).

Weakness：
1.	The model employs MAP estimation with SGD restarts rather than full Bayesian sampling. This point estimation approach may affect the accuracy of uncertainty quantification (e.g., the stability of PIPs) depending on initialization and optimization paths.
2.	The interpretability of the model is highly dependent on the number of spline bases J_n. While J_n determines the accuracy of the functional approximation , it also directly controls the granularity of region selection. In the ECG experiment, the chosen values for J_n (up to 220) are very close to the number of original discrete sampling points (256). Under such high-resolution settings, the identified "regions" may degenerate into extremely narrow temporal spikes. This makes the approach functionally equivalent to discrete variable selection, potentially undermining the paper's core motivation to leverage the "continuous and structured nature" of functional data for region-level selection . The paper lacks a discussion on this risk of "high-resolution-induced degeneration" and does not analyze how increasing J_n affects the physical continuity and meaningfulness of the identified regions.
3.	Although the introduction of a deep neural network enables the model to capture nonlinear interactions among region-level features, the functional embedding within each region still relies on a linear integral operator int_{R_k}\beta_k(t)X(t)dt. This means that within each local region, the influence of the functional predictor on the response is restricted to a single linear projection direction. In certain practical applications (e.g., neuroimaging analysis), functional signals within the same temporal/spatial region may influence the response through multiple independent feature directions (i.e., a within-region multi-index structure). For instance, the mean level and oscillation frequency of an fMRI signal segment may carry distinct diagnostic information, but the current framework can only extract a single linear projection feature per region. Have the authors considered extending the within-region embedding to multi-dimensional projections (e.g., using multiple basis function projection directions per region) to enhance the capacity for capturing complex functional features within regions? What impact would such an extension have on the theoretical results (particularly the approximation error bounds and posterior contraction rates)?

---

> ### Author Rebuttal · Authors · 2026-03-31
>
> Thank you for your thorough and constructive review. We appreciate your recognition of our work’s strengths. We have addressed the main concerns by: justifying MAP‑based PIPs with stability experiments; analyzing the effect of $J_n$ and adding sensitivity studies; discussing multi‑index extensions; adding weaker‑signal simulations; and reporting computational times. Due to space constraints, detailed results (Responses 1, 2, 5, 7) are available at https://osf.io/3gehx/overview?view_only=af6c51d11d304c58aa8210657e58a04f.
>
> 1. Weakness 1: While full Bayesian sampling is more common for uncertainty quantification, MAP‑based PIPs are a well‑established approach in sparse Bayesian modeling (Ročková, 2018 AOS; Ročková and George, 2018 JASA; Gan et al., 2019 JASA; Yang et al., 2021 JMLR), offering theoretical guarantees and computational efficiency. We have conducted additional experiments showing that PIPs and region selections are stable across different initializations. We will add a discussion of these points in Conclusion and include the stability experiments in the revised manuscript.
>
> 2. Weakness 2: We acknowledge that $J_n$ affects region selection granularity. With fixed‑order B‑splines, each basis function has local support of order $1/J_n$ and overlaps with neighbors, so selected regions remain unions of intervals rather than degenerating into pointwise variables. In ECG, a larger $J_n$ was used because the signal contains a narrow peak; smaller $J_n$ would yield overly broad selections. Sensitivity analyses across simulation and ECG data further show that a larger $J_n$ increases resolution, enabling detection of narrow intervals that can later be merged into larger meaningful regions, confirming that higher $J_n$ does not simply lead to smaller regions, but preserves interval-based structure. We acknowledge that larger $J_n$ reduces smoothness to some extent, but the effect is not severe. As described in Algorithm 1, we select $J_n$ from a candidate grid using a Laplace-type evidence surrogate. We will add a discussion and include the experimental results in the revised manuscript.
>
> 3. Weakness 3: The extension to within‑region multi‑index embeddings is natural: one can use $M$ basis projections per region and stack them as inputs to the DNN. If $M$ remains bounded (or grows in a controlled way), the current approximation and posterior‑contraction framework should be extendable in spirit, though the rates would reflect the higher effective dimension. The main challenge is identifiability: the meaningful object becomes a region‑wise projection subspace rather than a single direction, introducing rotational non‑identifiability and complicating the definition of an active region. Stable estimation would likely require additional constraints such as orthogonality or subspace‑level sparsity. We will add a discussion of this extension in Conclusion.
>
> 4. Q1: Please see the response to Weakness 2.
>
> 5. Q2: The minimum signal strength condition is standard in high‑dimensional variable and region selection literature, typically stated with an unspecified constant and rarely verified empirically—a limitation shared by most theoretical work. Our original experiments considered SNR levels of 5 and 10. To further address your concern, we conducted additional simulations with weaker signals. As expected, all methods become more challenging, yet our approach remains stable and continues to outperform the baselines. We will add these results and discussions to the revised manuscript.
>
> 6. Q3: We agree that the lower precision indicates the selected regions are somewhat wider than the predefined QRS silver interval. However, we do not interpret this as a localization failure. The silver interval is a simplified fixed proxy, whereas the predictive morphology for QRS duration may extend into adjacent transition regions that carry clinically relevant information. Precision computed against such a narrow reference is therefore conservative. We will clarify this point in the manuscript to avoid overinterpreting the ECG results as exact support recovery.
>
> 7. Q4: We have compared running times across sample sizes $n$ and numbers of observed time points. Even with multiple initializations, our method is computationally efficient. For IHPC, the entire training-and-selection procedure over all three candidate $J_n$ values completes within about 12 minutes. We will include these results in the revised manuscript.
>
> 8. Limitations: We will expand the Conclusion to discuss three limitations: (1) dependence on B-spline configuration (we have investigated the effect of the number of bases and will consider adaptive selection in future work); (2) current focus on univariate functional inputs, with extension to multi-source and collinear settings as future work using group-sparse priors; and (3) clinical implications, emphasizing that results should be interpreted with domain expertise and caution.

---

> > ### Author Rebuttal · Reviewer_xutz · 2026-04-01
> >
> > Most of my concerns are addressed. I am happy to raise the score to weak accept.

---

> > > ### Author Response · Authors · 2026-04-01
> > >
> > > We sincerely thank you for the thoughtful and constructive feedback. We are glad that our clarifications and additional analyses addressed your main concerns, and we greatly appreciate the updated evaluation.

---

### Decision · Program_Chairs · 2026-04-30

**Decision:**

Accept (regular)

**Comment:**

- The reviews converged positively after the rebuttal: reviewers agreed that the paper addresses an important problem in functional data analysis by combining nonlinear prediction with interpretable region selection, and they viewed the methodological combination and theoretical development as meaningful strengths.
- The main initial concerns were about the use of MAP-based rather than full Bayesian inference, the dependence on spline resolution for interpretability, theoretical/model alignment, and identifiability of selected regions, but the rebuttal addressed these through clarifications, additional stability and sensitivity analyses, expanded discussion of limitations, and stronger explanation of how the theory applies to the fitted model.
- Overall, the discussion shifted toward acceptance, with the remaining issues framed as limitations and clarifications for the final version rather than fundamental flaws.